# Ion acoustic waves near a comet nucleus: Rosetta observations at comet 67P/Churyumov-Gerasimenko

Herbert Gunell[1], Charlotte Goetz[2], Elias Odelstad[3], Arnaud Beth[1], Maria Hamrin[1], Pierre Henri[4,5], Fredrik L. Johansson[6], Hans Nilsson[7], and Gabriella Stenberg Wieser[7]

[1]Department of Physics, Umeå University, 901 87 Umeå, Sweden
[2]Space Research and Technology Centre, European Space Agency, Keplerlaan 1, 2201AZ Noordwijk, The Netherlands
[3]Department of Space and Plasma Physics, Royal Institute of Technology, 100 44 Stockholm, Sweden
[4]LPC2E, CNRS, F-45071 Orléans, France
[5]Lagrange, OCA, CNRS, UCA, Nice, France
[6]Swedish Institute of Space Physics, Box 537, 751 21 Uppsala, Sweden
[7]Swedish Institute of Space Physics, Box 812, 981 28 Kiruna, Sweden

**Correspondence:** Herbert Gunell (herbert.gunell@physics.org)

**Abstract.** Ion acoustic waves were observed between 15 and 30 km from the centre of comet 67P/Churyumov-Gerasimenko by the Rosetta spacecraft during its close flyby on 28 March 2015. There are two electron populations: one cold at $k_B T_e \approx 0.2\,\mathrm{eV}$ and one warm at $k_B T_e \approx 2\,\mathrm{eV}$. The ions are dominated by a cold (a few hundredths of eV) distribution of water group ions with a bulk speed of $(3\text{–}3.7)\,\mathrm{km\,s^{-1}}$. A warm $k_B T_e \approx 6\,\mathrm{eV}$ ion population, which also is present, has no influence on the ion
5  acoustic waves due to its low density of only 0.25 % of the plasma density. Near closest approach the propagation direction was within $50°$ from the direction of the bulk velocity. The waves, which in the plasma frame appear below the ion plasma frequency $f_{pi} \approx 2\,\mathrm{kHz}$, are Doppler shifted to the spacecraft frame where they cover a frequency range up to approximately $4\,\mathrm{kHz}$. The waves are detected in a region of space where the magnetic field is piled up and draped around the inner part of the ionised coma. Estimates of the current associated with the magnetic field gradient as observed by Rosetta are used as input to
10  calculations of dispersion relations for current-driven ion acoustic waves, using kinetic theory. Agreement between theory and observations is obtained for electron and ion distributions with the properties described above. The wave power decreases over cometocentric distances from 24 to 30 km. The main difference between the plasma at closest approach and in the region where the waves are decaying is the absence of a significant current in the latter. Wave observations and theory combined supplement the particle measurements that are difficult at low energies and complicated by spacecraft charging.

## 1 Introduction

Observations of waves can give us information of the plasma in which they are generated and through which they have propagated. Waves are also of general interest in plasma physics as they provide a means for energy transfer and because they

affect the charged particle distributions through wave–particle interaction processes. When comets 21P/Giacobini-Zinner and 1P/Halley were visited by spacecraft in the 1980s a variety of plasma waves were reported (Scarf et al., 1986c, a; Scarf, 1989). Among these observations were ion acoustic waves, detected both in the bow shock region (Scarf et al., 1986b) and upstream (Oya et al., 1986).

The Rosetta spacecraft (Glassmeier et al., 2007a) accompanied comet 67P/Churyumov-Gerasimenko for two years from August 2014 to September 2016. Shortly after the spacecraft reached the comet, low frequency ($f \lesssim 100\,\mathrm{mHz}$) long wavelength ($100\,\mathrm{km} \lesssim \lambda \lesssim 700\,\mathrm{km}$) waves were detected in the magnetic field data (Richter et al., 2015, 2016). These were named "singing comet" waves; they have been interpreted in terms of a modified ion-Weibel instability (Meier et al., 2016), found to be compressional (Breuillard et al., 2019), and detected as far as 800 km from the nucleus (Goetz et al., 2020). Waves in the lower hybrid frequency range ($f \lesssim 15\,\mathrm{Hz}$) were found by André et al. (2017) and Karlsson et al. (2017). Lower hybrid waves were frequently seen in bursts in connection with density gradients, oscillating on minute time scales (Stenberg Wieser et al., 2017). These minute time scale oscillations are known as steepened waves and were observed outside the diamagnetic cavity (Goetz et al., 2016b, a). Electric field measurements showed waves in the lower hybrid frequency range on both sides of the diamagnetic cavity boundary, indicating a mode conversion between lower hybrid waves and ion acoustic waves (Madsen et al., 2018).

Ion acoustic waves are compressional plasma waves that are weakly damped only when $T_\mathrm{e} \gg T_\mathrm{i}$, where $T_\mathrm{e}$ and $T_\mathrm{i}$ are the electron and ion temperatures respectively, and the frequency is below the ion plasma frequency. In this limit, the angular frequency $\omega$ is proportional to the wave number $k$ and the phase speed is $c_\mathrm{s} = \sqrt{k_\mathrm{B}T_\mathrm{e}/m_\mathrm{i}}$, where $k_\mathrm{B}$ is Boltzmann's constant and $m_\mathrm{i}$ is the ion mass (see for example Krall and Trivelpiece, 1973). As the frequency approaches the ion plasma frequency they become increasingly heavily damped and also the phase speed decreases. If the ion and electron temperatures are similar, this also leads to heavy damping, and ion acoustic waves are usually not detectable in that regime. Charged particle disributions become unstable when one population drifts at a large enough speed relative another population, and this results in the growth of waves. In particular, this applies to the current-driven ion acoustic instability, where the electron and ion populations are in relative motion. The current-driven ion acoustic instability has been studied in laboratory experiments (Sato et al., 1976; Kawai et al., 1978; Michelsen et al., 1979), and Stringer (1964) mapped out the unstable parameter regimes theoretically. At comet 67P/Churyumuov-Gerasimenko ion acoustic waves were observed by the Rosetta spacecraft on 20 January 2015 (Gunell et al., 2017b) at approximately 2.5 AU from the Sun, before the diamagnetic cavity had formed (see Goetz et al., 2016a, for a discussion of diamagnetic cavity formation), and also in the diamagnetic cavity near perihelion (Gunell et al., 2017a). The ion acoustic waves seen in the cavity were interpreted as a result of part of the current at the diamagnetic cavity boundary closing through bulges on that boundary and generating waves through a current–driven instability (Gunell et al., 2017a). Ion acoustic waves can be identified by measuring either the variations in plasma density or in the electric field. At comet 67P/Churyumov-Gerasimenko, Gunell et al. (2017b) detected the waves in electric field oscillations and Gunell et al. (2017a) in density variations. In this work, the detection relies on density variations.

In this article, we examine ion acoustic waves detected by Rosetta during its close flyby of comet 67P on 28 March 2015. The comet was at a heliocentric distance of 2.0 AU at the time, and the gas production rate varied between $3 \times 10^{26}\,\mathrm{s^{-1}}$ and

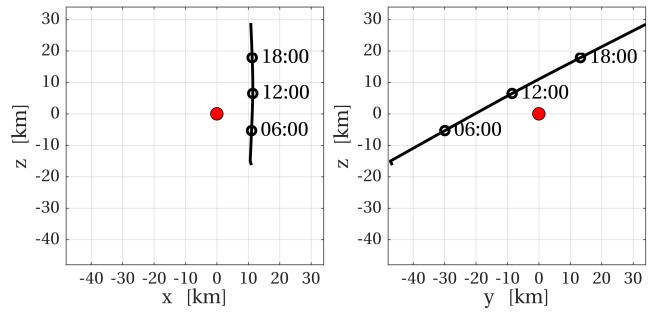

**Figure 1.** Trajectory followed by the Rosetta spacecraft during the close flyby on 28 March 2015. The red circle represents the nucleus of comet 67P/Churyumov-Gerasimenko.

$9 \times 10^{26}\,\mathrm{s}^{-1}$ during the day. Magnetic pileup and draping have been studied before for this flyby, both in observational studies and using hybrid simulations (Koenders et al., 2016). The magnetic field piled up near the nucleus, causing the solar wind protons to be deflected out of the ecliptic plane. This in turn caused the draped magnetic field in the region near the nucleus to align itself with the deflected solar wind flow. No sign of a diamagnetic cavity was seen during the flyby, and that was likely due to it not having formed yet (Goetz et al., 2016a). The hybrid simulations of the flyby presented by Koenders et al. (2016) show the presence of an infant bow shock (Gunell et al., 2018) approximately $100\,\mathrm{km}$ from the nucleus, but that is farther out than the spacecraft reached on that day. Thus, the spacecraft was situated in the inner coma, where the plasma was of cometary origin. The plasma parameters, although not exactly the same, were in a range similar to that of the diamagnetic cavity observations near perihelion. The magnetic field environment was dominated by magnetic pileup and draping during the flyby, while the diamagnetic cavity has its own peculiar magnetic field environment with a sharp discontinuity at the boundary.

## 2 Observations

We use the comet-centred solar equatorial coordinate system (CSEQ) throughout this article. In this system, the $x$ axis points from the comet to the Sun, the $z$ axis is the component of the rotation axis of the Sun that is perpendicular to the $x$ axis, and the $y$ axis is directed to complete the right-handed coordinate system (original definition in the SPICE kernel, Acton, 1996). The spacecraft moved from negative to positive $y$ and $z$ values at a nearly constant $x = 11\,\mathrm{km}$. The spacecraft trajectory is illustrated in Fig. 1.

The closest approach occured at 13:05 UTC, and then Rosetta was at a cometocentric distance of $15\,\mathrm{km}$. The spacecraft moved slowly (with a relative speed to the comet below $1\,\mathrm{m\,s}^{-1}$) and was in the the vicinity of the nucleus for several hours as shown in Fig. 1 and in panel h of Fig. 2.

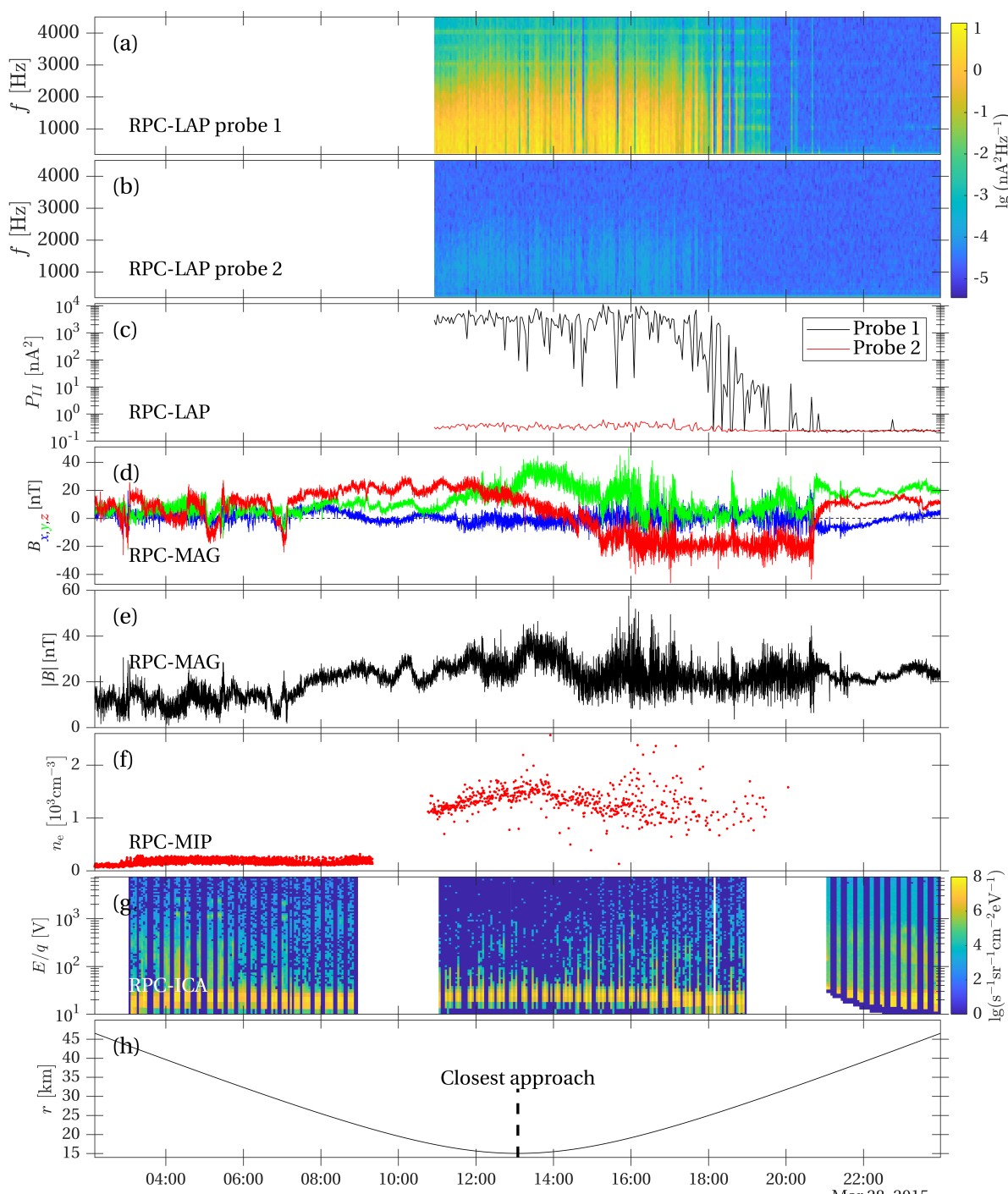

**Figure 2.** Rosetta observations during the close flyby of comet 67P/Churyumov-Gerasimenko on 28 March 2015. **(a)** Power spectral density of RPC-LAP probe 1 in the frequency range $200\,\text{Hz} < f < 4.5\,\text{kHz}$. **(b)** Power spectral density of RPC-LAP probe 2 in the same frequency range. **(c)** The power spectral density of probes 1 and 2 integrated from $200\,\text{Hz}$ to the Nyquist frequency of $9375\,\text{Hz}$. RPC-LAP probe 1 was biased to $+30\,\text{V}$ and probe 2 was biased to $-30\,\text{V}$ with respect to the spacecraft potential. **(d)** $B_x$ (blue), $B_y$ (green), and $B_z$ (red) components of the magnetic flux density measured by RPC-MAG. **(e)** The magnitude of the magnetic flux density. **(f)** The plasma density derived from RPC-MIP data. **(g)** Ion energy spectrum observed by RPC-ICA summed over all angles and mass channels. **(h)** Cometocentric distance of the Rosetta spacecraft.

## 2.1 Instrumentation

The data used in this article was obtained by instruments belonging to the Rosetta Plasma Consortium (RPC) (Carr et al.,
2007). For the wave observations (Sect. 2.2) we used the Rosetta Langmuir probe instrument (RPC-LAP) (Eriksson et al.,
2007) to record time series of probe current variations attributed to waves in the cometary plasma environment. RPC-LAP
is constituted of two spherical probes, 5 cm in diameter, that are mounted on booms protruding from the spacecraft. Starting
at 10:55:34 on 28 March 2015, RPC-LAP regularly recorded such time series for the rest of the day. The probe current was
sampled at a frequency of $f_\mathrm{s} = 18750\,\mathrm{Hz}$, and each time series contains 1600 samples, corresponding to a time series length
of 85.3 ms. This process was repeated every 160 s. Each of the two probes obtained 295 such time series during the day. The
power spectral density for each time series is computed, using Welch's method (Welch, 1967), averaging segments that are 256
samples long with an overlap of 65 % (Fig. 2a and b). The probes were held at fixed potentials with respect to the spacecraft:
probe 1 at $+30\,\mathrm{V}$ and probe 2 at $-30\,\mathrm{V}$. For reference, the spacecraft potential was approximately $-20\,\mathrm{V}$ with respect to the
plasma. The Langmuir probe instrument was also used to derive the bulk speed of the ions and the electron temperature by
sweeping the probe potential and measuring the probe current as described in Sect. 2.3.

We use the Mutual Impedance Probe (RPC-MIP) (Trotignon et al., 2007) to obtain the plasma density during the flyby. The
RPC-MIP instrument observes the plasma frequency, from which the plasma density is derived (Fig. 2f). The ion populations
are sampled by the Ion Composition Analyser (RPC-ICA) (Nilsson et al., 2007) (Fig. 2g). The magnetic field is measured by
the magnetometer (RPC-MAG) (Glassmeier et al., 2007b). The magnetic field components are presented in CSEQ coordinates
in Fig. 2d. How the properties of the plasma are derived from the data collected by these instruments is described in Sect. 2.3.

## 2.2 Waves

Power spectral densities obtained for RPC-LAP probe 1 are shown in Fig. 2a, and those recorded by probe 2 are shown in
Fig. 2b. The colour coded quantity is the logarithm of the power spectral density (PSD) of the probe currents. The lowest
frequency bins are at risk of picking up low-frequency noise, and we therefore show the spectrum for frequencies above
200 Hz. There may be other waves present at low frequencies, but in this article we only consider ion acoustic waves above
200 Hz. The waves are identified as ion acoustic waves, because they are compressional, showing a plasma density variation,
and other wave modes can be excluded for the spatial and temporal scales where they are observed as shown in Sect. 2.4.

A high amplitude wave signal is seen during the close flyby and it falls off as the spacecraft moves away from the nucleus.
The power spectral density of the positively biased probe 1 is several orders of magnitude higher than that of the negative
probe 2. This means that the probe 1 signal is dominated by the electron current and that the signal is proportional to the
density variation of the wave. The probe operated in this regime also in the previously published observations of waves in the
diamagnetic cavity (Gunell et al., 2017a), whereas in the first published observations (Gunell et al., 2017b), the probe was
capacitively coupled to the plasma. A simple test to distinguish between the two regimes is to measure the wave amplitude as
a function of probe bias, where a negatively biased probe suppresses the electron current (Torvén et al., 1995). In the present
case, we compare the two probes that are biased differently (see also Gunell et al., 2017b). Also the maximum PSD value is

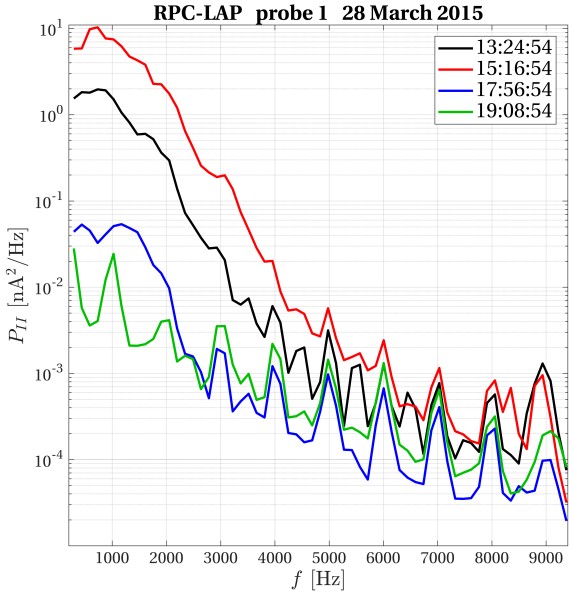

**Figure 3.** Power spectral densities from 200 Hz up to the Nyquist frequency for the RPC-LAP probe 1 current for four different times during the Rosetta close flyby of comet 67P. The black curve (13:24:54) is used in Sect. 3 for analysis of waves near closest approach and the blue curve (17:56:54) for similar analysis of during the outbound part of the flyby.

similar to those observations and the plasma density, shown here in Fig. 2f, was in both cases somewhat above $1000\,\mathrm{cm}^{-3}$. The situation differs from the first ion acoustic wave observations at comet 67P (Gunell et al., 2017b) when the plasma density was an order of magnitude smaller and the waves coupled capacitively to the probe through the displacement current instead of a particle current. The difference between the probes is also seen in Fig. 2c, which shows the integral of the power spectral
density over frequencies from 200 Hz up to the Nyquist frequency.

Fig. 3 shows four sample spectra of the probe 1 current for frequencies from 200 Hz up to the Nyquist frequency. There is wave power starting at the low end of this frequency range with a broad maximum in the vicinity of 1 kHz, and at higher frequencies the PSD declines toward the noise floor. The black curve shows the PSD at 13:24:54, which is near closest approach to the comet nucleus at a cometocentric distance of 15 km. As seen in Fig. 2c the total wave power fluctuated but remained at
a generally high level while the spacecraft was in the near-nucleus environment. The PSD obtained at 15:16:54 (red curve in Fig. 3) is another example from this period. The spacecraft was at 17.5 km cometocentric distance and the wave power was even higher than that shown by the black curve. The wave power declined as the spacecraft moved to larger cometocentric distances. This process started approximately at 17:45 when Rosetta was at 24 km from the centre of the nucleus. Two examples from the declining phase are shown in Fig. 3: the spectrum obtained at 17:56:54 at 25 km (blue curve) and one spectrum from 19:08:54
at 29 km (green curve) when the wave power had fallen even more. The two curves that will be used for comparison with wave theory in Sect. 3 are the black curve (13:24:54) for closest approach and the blue curve (17:56:54) for the outbound case. The

peaks at multiples of 1 kHz seen in the frequency range where the wave power is low, both in Fig. 3 and Fig. 2 are artefacts generated by the spacecraft.

## 2.3 Plasma properties

To analyse the waves we need to know the basic properties of the plasma. The plasma density obtained by the mutual impedance probe, RPC-MIP, is shown in Fig. 2f. The density peaks around closest approach and then falls off as the spacecraft moves away from the nucleus. The scattered instantaneous plasma density values are a signature of strong plasma inhomogeneities of approximately 10 % around closest approach. For the calculations in Sect. 3 we estimate a plasma density of $n_e = 1600 \, \text{cm}^{-3}$ at closest approach and $n_e = 1000 \, \text{cm}^{-3}$ for the outbound case.

Fig. 2g shows an energy spectrum of the positive ions observed by RPC-ICA. At $E/q \approx 20 \, \text{V}$ is a warm ($k_B T_i \approx 6 \, \text{eV}$ around the time of closest approach) water ion population, which has been accelerated toward the spacecraft due to the negative spacecraft potential. The temperature was obtained by fitting a drifting Maxwellian to the data recorded by the instrument. Some accelerated water ions are seen at higher energies, but the vast majority of the ions seen in Fig. 2g belong to the warm, low energy, population. Fitting the observed flux to a drifting Maxwellian distribution we arrive at a density estimate of about

$4 \, \text{cm}^{-3}$ for this ion population. However, this is far below the $1600 \, \text{cm}^{-3}$ plasma density measured by RPC-MIP. The sensitivity of RPC-CIA is low in the lowest energy range, and the angular range that allows entry into the instrument is narrow. A monoenergetic low energy beam is liable to be undetected. Thus, the discrepancy between the RPC-ICA measured ion density and the plasma density measured by RPC-MIP is explained by a cold water ion distribution that is invisible to RPC-ICA.

This is confirmed by Langmuir probe characteristics shown in Fig. 4. Fig. 4b shows the part of the characteristics dominated

by the ion current. In the following we estimate the bulk speed of the cold ions. The warm ion population is negligible because of its low density, and cold plasma theory is applicable because the thermal speed of the cold ions is far below their bulk speed. For a cold ion population drifting at a bulk speed $u$ the probe current $I$ depends on the probe to plasma potential $V$ according to (Mott-Smith and Langmuir, 1926)

$$I = -\pi r_p^2 n_i u e \left(1 - \frac{2eV}{m_i u^2}\right), \tag{1}$$

where $r_p = 2.5 \, \text{cm}$ is the radius of the probe, $m_i$ is the ion mass and $n_i$ the ion density. We fit a line to the linear part of the curve, and taking the derivative of Eq. (1) and rearranging we can determine the drift velocity from the slope $dI/dV$ of that line:

$$u = \frac{2 n_i e^2 \pi r_p^2}{m_i \frac{dI}{dV}}. \tag{2}$$

Taking the ion density to be equal to the plasma density measured by RPC-MIP we arrive at an ion drift speed of $3 \, \text{km} \, \text{s}^{-1}$ near

closest approach and $3.7 \, \text{km} \, \text{s}^{-1}$ at 17:52:06 when the spacecraft was moving outward as shown in Fig. 4b. These numbers are within the range of those observed by Odelstad et al. (2018) in spite of the differences in the magnetic field environment and distance to the nucleus. The ion temperature can be estimated from the neutral temperature, as ions are created by ionisation of the neutrals. Biver et al. (2019) found neutral temperatures in the 50–200 K range, which corresponds to approximately $0.02 \, \text{eV}$,

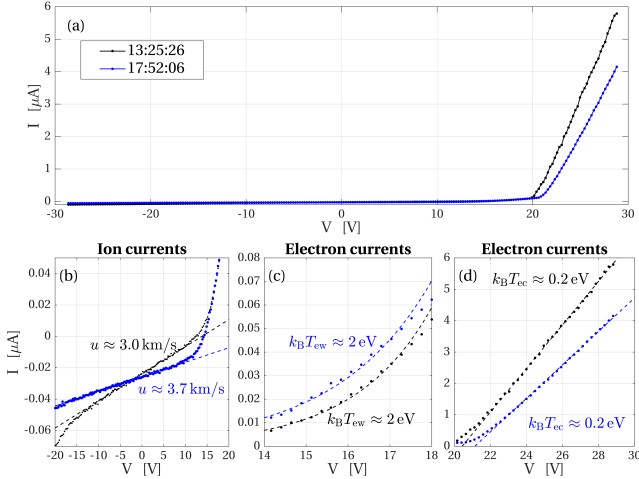

**Figure 4.** $I$–$V$ traces from RPC-LAP near closest approach (black) and during the outbound part of the flyby (blue). **(a)** complete $I$–$V$ traces. **(b)** ion current part of the characteristics and lines fitted to the ion currents in order to find the bulk speed of the cold ion population. **(c)** electron currents and exponentials fitted to determine the warm electron temperature. **(d)** electron saturation current part of the characteristics and lines fitted to determine the temperature of the cold electron population.

and that is well below the $1\,\mathrm{eV}$ kinetic energy, corresponding to the $3$–$3.7\,\mathrm{km\,s^{-1}}$ drift speeds obtained above. This confirms
the assumptions stated above Eq. (1) and ensures the applicability of the equation. Photo emission comes in as an offset in the ion current, below the plasma potential, and as such, does not play a role in the slope of the ion current. Furthermore, for the first of the sweeps, the probe was in shadow behind the spacecraft. The deviation from the linear fit at the low voltage end of Fig. 4b could be caused by secondary emission provoked by impacting ions, but the quantum yield is low for low energies and our analysis of the ion current only starts at -10 V. For the wave analysis, the probe is positively biased, rendering ion impact,
and hence also secondary emission, negligible.

Fig. 4d shows the part of the probe characteristic where the current is dominated by the electrons. The dashed lines have been fitted to the high probe potential part of the sweep. Here, the current varies linearly with voltage (Swift and Schwar, 1970) and the cold electron temperature is (Engelhardt et al., 2018)

$$T_{\mathrm{e}} = 8\pi \frac{r_{\mathrm{p}}^4 e^3 n_{\mathrm{e}}^2}{m_{\mathrm{e}}} \left( \frac{\mathrm{d}I}{\mathrm{d}V} \right)^{-2}. \tag{3}$$

The slopes of these lines correspond to temperatures of $k_{\mathrm{B}}T_{\mathrm{e}} = 0.2\,\mathrm{eV}$ for both the closest approach and outbound curves. The curves also show that the plasma potential is approximately 20 volts above the spacecraft potential. It was shown in simulations by Johansson et al. (2020) that the spacecraft potential is driven negative by positively biased elements on the solar panels that collect cold electrons from the plasma. The estimate we use in Sect. 3 is that the electron distribution is constituted by two contributions with equal densities: one cold with temperatures as estimated in Fig. 4d and one warm with a temperature of
$2\,\mathrm{eV}$. The latter is found by fitting exponential curves to the part of the $I$–$V$ trace dominated by the warm electrons as shown in Fig. 4c. These results are similar to previous Langmuir probe sweep interpretations from when the comet was near perihelion

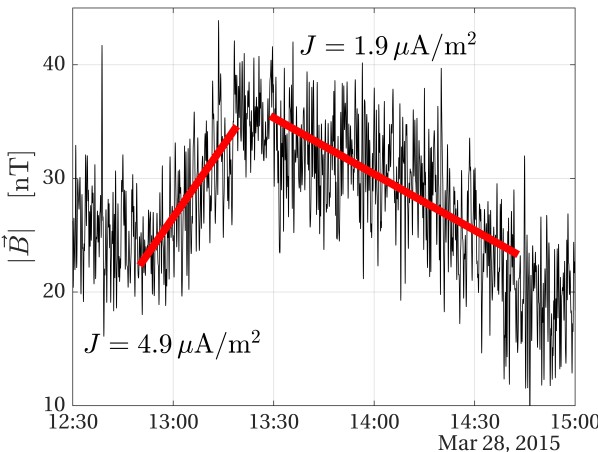

**Figure 5.** Magnetic field magnitude during a period around closest approach. The red lines are fitted to the data in order to derive the current densities $J = 4.9\,\mu\text{A m}^{-2}$ (inbound) and $J = 1.9\,\mu\text{A m}^{-2}$ (outbound).

(Eriksson et al., 2017; Gunell et al., 2017a; Odelstad et al., 2018) with the difference that the cold electrons are not quite as cold here as the 0.1 eV that was estimated near perihelion. These two electron temperature values are within the range of those observed by RPC-MIP at similar heliocentric distances in 2016 (Wattieaux et al., 2020). The general instrumental response of

175 the RPC-MIP mutual impedance probe in a plasma characterised by two electron populations with different temperatures is described by Gilet et al. (2017) and Wattieaux et al. (2020).

Figure 2d shows the components and Fig. 2e the magnitude of **B** as measured by RPC-MAG. The magnitude of the magnetic field increased as the spacecraft approached the centre of the comet and decreased as it was moving away. This is expected from magnetic pileup and field line draping, but there are also other changes in the magnetic field that can be seen in Fig. 2d

and e.

To estimate the current associated with the non-uniformity of the magnetic field we fit lines to the magnitude of the magnetic field as shown in Fig. 5. Assuming that the spacecraft moves through a stationary magnetic field, we estimate the magnitude of the current density by

$$J = \frac{|\Delta B|}{\mu_0 |\Delta \mathbf{r}|}. \tag{4}$$

where $|\Delta B|$ is the change in the fitted magnetic field magnitude and $|\Delta \mathbf{r}|$ is the distance the spacecraft moved during the same period of time. This yields a current density of $J = 4.9\,\mu\text{A m}^{-2}$ when the spacecraft was approaching the nucleus and $J = 1.9\,\mu\text{A m}^{-2}$ while it was moving away. These values should be seen as estimates of the average current density. Koenders et al. (2016) compared the magnetic field observed during this flyby to the magnetic field obtained in hybrid simulations and found good agreement for $B_y$, which is the dominating component. The simulated $B_z$ component was about a factor of 2

lower than what was observed by Rosetta. The difference could be attributed to the limited resolution or the use of an averaged

outgassing profile in the simulations (Koenders et al., 2016). For the magnitude of **B** the difference only amounts to 10–20 %, but the simulation does not follow how the plasma quantities develop in time. Fig. 5 shows that the magnetic field changed on much shorter timescales than those of our linear approximations during the flyby. From a single spacecraft measurement we cannot determine whether these magnetic field fluctuations are due to local variations of the current in the plasma or whether the whole inner region of the ionised coma is undergoing oscillations. Thus, the current density may have been both higher and lower than these average values during the flyby.

## 2.4  Typical scales

A plasma density of $1600 \, \mathrm{cm}^{-3}$ corresponds to an electron plasma frequency of approximately $350 \, \mathrm{kHz}$ and a $H_2O^+$ ion plasma frequency of $2 \, \mathrm{kHz}$. Thus, electron time scale waves, such as Langmuir waves and electron acoustic waves, are far beyond reach of our observations, the Langmuir probe being sampled at the much lower frequency of $18.75 \, \mathrm{kHz}$. Ion acoustic waves, on the other hand, are in the accessible frequency range. The magnetic field during the flyby varied between 20 and $40 \, \mathrm{nT}$ approximately. This corresponds to electron cyclotron frequencies between 0.6 and $1.1 \, \mathrm{kHz}$, which is in the middle of the observed frequency range. However, the observed wave frequency does not follow the changes in the magnetic field, which rules out electron cyclotron waves. For example, the PSD peaks at $700 \, \mathrm{Hz}$ for both times 13:24:54 and 15:16:54, shown by the black and red curves in Fig. 3, respectively, even though the electron cyclotron frequency was $1.1 \, \mathrm{kHz}$ at 13:24:54 and $0.57 \, \mathrm{kHz}$ at 15:16:54. The ion cyclotron frequency is $(0.02 - 0.03) \, \mathrm{Hz}$, which is below the frequencies we can resolve.

The spacecraft was at $15 \, \mathrm{km}$ cometocentric distance at closest approach and at $25 \, \mathrm{km}$ at 18:00 when the wave amplitude started to decrease. Thus, the typical length for the variation in wave amplitude is about $10 \, \mathrm{km}$. Assuming a typical $B$ of $30 \, \mathrm{nT}$ warm ions at $6 \, \mathrm{eV}$ would have a gyroradius of $50 \, \mathrm{km}$. Cold ions are picked up by the electric field, moving along trajectories with a radius of curvature that is even larger. The ions can thus be seen as unmagnetised. Warm electrons at $4 \, \mathrm{eV}$ have gyroradius of $225 \, \mathrm{m}$ and for cold $0.2 \, \mathrm{eV}$ electrons the gyroradius is $50 \, \mathrm{m}$ approximately.

In Sect. 3 we use kinetic theory to compute dispersion relations for electrostatic waves in an unmagnetised plasma. This is applicable if the wavelength is much shorter than the gyroradii of the electrons and ions so that the influence of magnetic forces on particle motion is negligible on wavelength scales. In Sect. 3 it is seen that the phase speed for ion acoustic waves is approximately $1.7 \, \mathrm{km \, s}^{-1}$. Thus, a wave at $200 \, \mathrm{Hz}$ (the lower limit of the spectrum shown in Fig. 2a) has a wavelength of $8.5 \, \mathrm{m}$, which is far below all the gyroradii reported above. The assumption that the plasma is unmagnetised for wave purposes holds above that limit, and these are the waves considered here. For waves at the very lowest frequencies, below the range considered here, the wavelength is longer, and electromagnetic effects would have to be taken into account.

## 2.5  Measurement uncertainties

All measurements are associated with uncertainties. The random error of the RPC-MIP derived densities are in the range of 10–20 %. The error is computed from the frequency resolution of the instrument, which measures the plasma frequency line of the mutual impedance spectra. For RPC-MAG the uncertainty in individual measurements is $5 \, \mathrm{nT}$ (Goetz et al., 2016a). This is less than the natural variations that are seen on the magnetic field curves in Fig. 2. The cold ion population is not detected

by RPC-ICA. Ions with energies above a few eV are detected and their energy is known to the precision of the width of the
energy bins, which is 30 % at energies below 30 eV. The direction from which the low energy ions arrive at the instrument
depends heavily on the electric fields around the spacecraft and it requires modelling to relate the observed arrival directions to
the travel directions of ions outside the spacecraft sphere of influence (Bergman et al., 2020a, b). For the very lowest energies
the sensitivity is low, and also ions can enter the instrument only from a narrow angular range. For RPC-LAP the uncertainty
lies more in the interpretation than in the measurement of currents and voltages. We will now discuss the uncertainties in those
interpretations that we have made, using data from all instruments.

The spacecraft potential is determined from the Langmuir probe characteristics, and that estimate could be off by $\sim 2$ eV.
This does not affect the slopes of the characteristics that are used to determine the electron cold and warm temperatures as well
as the ion drift speed. The temperature of the cold electrons is calculated using Eq. (3). The most significant contribution to
error in Eq. (3) is the electron density, which is squared in the equation. We used the density determined by RPC-MIP, which
has an error of 10–20 %. The total error in Eq. (3) is estimated to be $\lesssim 50$ %. The estimate of the warm electron temperature
could be influenced by other electron populations. However, the exponential fit ends 2 V below the plasma potential, and the
cold electrons have little influence there, given their low 0.2 eV temperature. The probe was in the shadow behind the spacecraft
during closest approach, which means that there was no influence from photo electrons. The measurement performed while
the spacecraft was outbound could in principle have been influenced by photo electrons, even though no substantial change in
temperature was observed. The possible error is estimated to be covered by the 1–4 eV range spanned by the test distributions
in Sect. 3. The ion velocity is found by the use of Eq. (2). The velocity is proportional to the density, which is obtained by
RPC-MIP, which means that the 10–20 % random error in those measurements applies to the velocity as well. Any influence
of a spacecraft sheath on the RPC-LAP measurements is negligible, since the probe is outside the sheath. The Debye length
in a plasma with a 2 eV electron temperature and a $1.6 \times 10^9$ m$^{-3}$ density is $\lambda_{De} = 26$ cm, which is much less than the boom
length of 2.24 m. This value of $\lambda_{De}$ is an upper limit, since it is based on the warm electrons. If the cold electrons would also
be included, lower values would be obtained, which would place the probe even farther outside the sheath. Deviations from the
nominal value due to density and temperature fluctuations are estimated at less than a factor of 2, and the probe would remain
outside the sheath also under such conditions.

The estimate of the warm ion temperature from the RPC-ICA observations is subject to an uncertainty of the order of the
spacecraft potential uncertainty $\sim 2$ eV. The cold ion temperature cannot be measured directly, but it can be confined to below
$\sim 1$ eV, since if the temperature were higher, the high energy tails would be visible in the energy range that can be observed.
The wave observations constrain the cold ion temperature further as is seen in the growth rate calculations with different
temperature in Sect. 3.

The current density estimate is influenced by the error in the magnetic field measurements and the error in the spacecraft
position. The latter can be neglected as it is of the order of tens of metres, and the spacecraft moved $|\Delta \mathbf{r}| = 2$ km and 5 km
during inbound and outbound measurement periods, respecively, as shown in Fig. 5. An upper limit of the error of the current
density estimate of 40 % is obtained if the full 5 nT random error of the individual estimate is applied to $|\Delta B|$ in Eq. (4).

**Table 1.** Parameters of the distributions used in the examples related to the plasma at closest approach. In the table, $n$ means density, $k_{\mathrm{B}}$ is Boltzmann's constant, $T$ denotes temperature, and $v_{\mathrm{D}}$ represents drift speed.

| | cold ions | | | warm ions | | | cold electrons | | | warm electrons | | |
|---|---|---|---|---|---|---|---|---|---|---|---|---|
| distr. | $n$ | $k_{\mathrm{B}}T$ | $v_{\mathrm{D}}$ | $n$ | $k_{\mathrm{B}}T$ | $v_{\mathrm{D}}$ | $n$ | $k_{\mathrm{B}}T$ | $v_{\mathrm{D}}$ | $n$ | $k_{\mathrm{B}}T$ | $v_{\mathrm{D}}$ |
| | [cm$^{-3}$] | [eV] | [km s$^{-1}$] | [cm$^{-3}$] | [eV] | [km s$^{-1}$] | [cm$^{-3}$] | [eV] | [km s$^{-1}$] | [cm$^{-3}$] | [eV] | [km s$^{-1}$] |
| | Closest approach | | | | | | | | | | | |
| 1 | 1554 | 0.02 | 0 | 4 | 6 | 0 | 779 | 0.2 | 39.3 | 779 | 2 | 0 |
| 2 | 1558 | 0.02 | 0 | 0 | – | – | 779 | 0.2 | 39.3 | 779 | 2 | 0 |
| 3 | 1518 | 0.02 | 0 | 40 | 6 | 0 | 779 | 0.2 | 39.3 | 779 | 2 | 0 |
| 4 | 1558 | 0.02 | 0 | 0 | – | – | 779 | 0.2 | 0 | 779 | 2 | 39.3 |
| 5 | 1558 | 0.02 | 0 | 0 | – | – | 779 | 0.2 | 39.3 | 779 | 1 | 0 |
| 6 | 1558 | 0.01 | 0 | 0 | – | – | 779 | 0.2 | 39.3 | 779 | 2 | 0 |
| 7 | 1558 | 0.04 | 0 | 0 | – | – | 779 | 0.2 | 39.3 | 779 | 2 | 0 |
| 8 | 1558 | 0.02 | 0 | 0 | – | – | 779 | 0.2 | 39.3 | 779 | 4 | 0 |
| 9 | 1558 | 0.02 | 0 | 0 | – | – | 779 | 0.2 | 19.6 | 779 | 2 | 19.6 |
| | Outbound | | | | | | | | | | | |
| A | 1006 | 0.02 | 0 | 0 | – | – | 503 | 0.2 | 23.6 | 503 | 2 | 0 |
| B | 1006 | 0.02 | 0 | 0 | – | – | 503 | 0.2 | 0 | 503 | 2 | 0 |

## 3 Dispersion relations

Because of the uncertainty at which both electron and ion distribution functions are known, we have calculated dispersion relations under several different assumptions about these distributions. We then compare the results of the calculations with the wave observations in order both to arrive at an explanation for how the waves are generated and to put constraints on what we can say about the electron and ion distributions. The total distribution function is composed of a cold and a warm electron and a cold and a warm ion distribution. The parameters are shown in Table 1 for 9 test cases used to model the distribution near closest approach and 2 cases for the outbound trajectory. We use the simple pole expansion method to compute the dispersion relations (Löfgren and Gunell, 1997; Gunell and Skiff, 2001, 2002; Tjulin et al., 2000; Tjulin and André, 2002). In a comet context it was reviewed by Gunell et al. (2017b, also providing the computer code for the computations). Each component of the distribution function is modelled by an approximate Maxwellian,

$$M_m(v) = \left[1 + \frac{(v-v_{\mathrm{D}})^2}{2v_{\mathrm{t}}^2} + \ldots + \frac{1}{m!}\left(\frac{(v-v_{\mathrm{D}})^2}{2v_{\mathrm{t}}^2}\right)^m\right]^{-1}, \tag{5}$$

where $v$ is velocity, $v_{\mathrm{t}}$ is the thermal speed, $v_{\mathrm{D}}$ is the drift speed, and $m$ is the number of terms included in the expansion. The procedure used to calculate dispersion relations is briefly described in Appendix A. The expression inside the brackets of

Eq. (5) is the reciprocal of a Taylor expansion of

$$\exp\left((v - v_{\mathrm{D}})^2 / \left(2v_{\mathrm{t}}^2\right)\right).$$

As $m$ tends to infinity $M_m(v)$ approaches a Maxwellian, and for small values of $m$ the distributions have suprathermal tails. In the distributions in Table 1, $m = 3$ for the ions and $m = 5$ for the electrons. The influence of suprathermal tails is evaluated in Appendix B.

Fig. 6. shows dispersion relations for the 9 test cases that correspond to the observations near closest approach. We are assuming a real wave number $k$ and a complex angular frequency $\omega$. Panel (a) shows the real part of $\omega$, and panel (b) shows the damping rate $\gamma$. Negative values of $\gamma$ correspond to wave growth. Panel (c) shows the shaded rectangle in panel (b) in more detail. Several of the curves are so similar they fall on top of each other and are difficult to distinguish in the figure. In all cases the least damped or fastest growing mode is the ion acoustic mode and that is the one shown.

The density of the warm ion population is varied in distributions 1–3. In distribution 1 the warm ion density is $4\,\mathrm{cm}^{-3}$ as estimated in Sect. 2.3. In distribution 2 the warm ion density is assumed to be zero, and in distribution 3 the warm ion density is ten times higher than the estimate in Sect. 2.3. The real part of the dispersion relation is indistinguishable among the three cases, as seen in Fig. 6a. The damping rates in Fig. 6c are very close in the three cases, although it can be descried that distribution 3, with the highest warm ion density, has a slightly smaller growth rate than the other two. However, the difference is small, and we conclude that the warm ion population only has a negligible influence on the waves. Therefore, the warm ion density is set to zero in the rest of the distribution functions.

We have used the current density estimate, $J = 4.9\,\mathrm{\mu A\,m}^{-2}$, obtained in Sect. 2.3 for the inbound part of the flyby. The dispersion relations are computed in the rest frame of the ions and the current is modelled by assigning a drift velocity, $|\mathbf{v}_{\mathrm{D}}| = |J/(en)| = 39.3\,\mathrm{km\,s}^{-1}$, to one of the electron populations. Here, $n$ denotes the density of the electron population in question. In distribution 2 and distribution 4 that drift speed is given to the cold and warm electron distribution, respectively. For distribution 4 the ion acoustic mode is damped, while it is growing for distribution 2. We conclude that to drive the ion acoustic waves unstable the current cannot be carried by the warm electrons alone. In distribution 9 the current is carried by both electron populations, which each are given a drift velocity of $|\mathbf{v}_{\mathrm{D}}| = 19.6\,\mathrm{km\,s}^{-1}$. This yields a lower growth rate than for distribution 2, but the mode is still unstable for a range of $k$ values.

In distribution 5, the temperature of the warm electrons has been decreased to $1\,\mathrm{eV}$. This leads to a decreased growth rate compared to distribution 2, which has $2\,\mathrm{eV}$ warm electrons but otherwise is equal to distribution 5. In distribution 8 the electron temperature has been increased to $4\,\mathrm{eV}$ with all other parameters the same as in distributions 2 and 5. The growth rate for distribution 8 is higher than that for the other two distributions. However, in all three cases the waves are unstable over a similar wavelength range.

Distributions 6 and 7 have $0.01\,\mathrm{eV}$ and $0.04\,\mathrm{eV}$ cold ions, respectively, that is to say, in distribution 6 the ions are colder and in distribution 7 warmer than they are in the otherwise equal distribution 2. This affects the growth rate so that distributions with colder ions grow faster and over a wider $k$ range than distributions where the ions are warmer (Fig. 6c). Also the real part of $\omega$ is affected, as shown in Fig. 6a, but this is significant only for $k$ values larger than the $k$ which corresponds to maximum

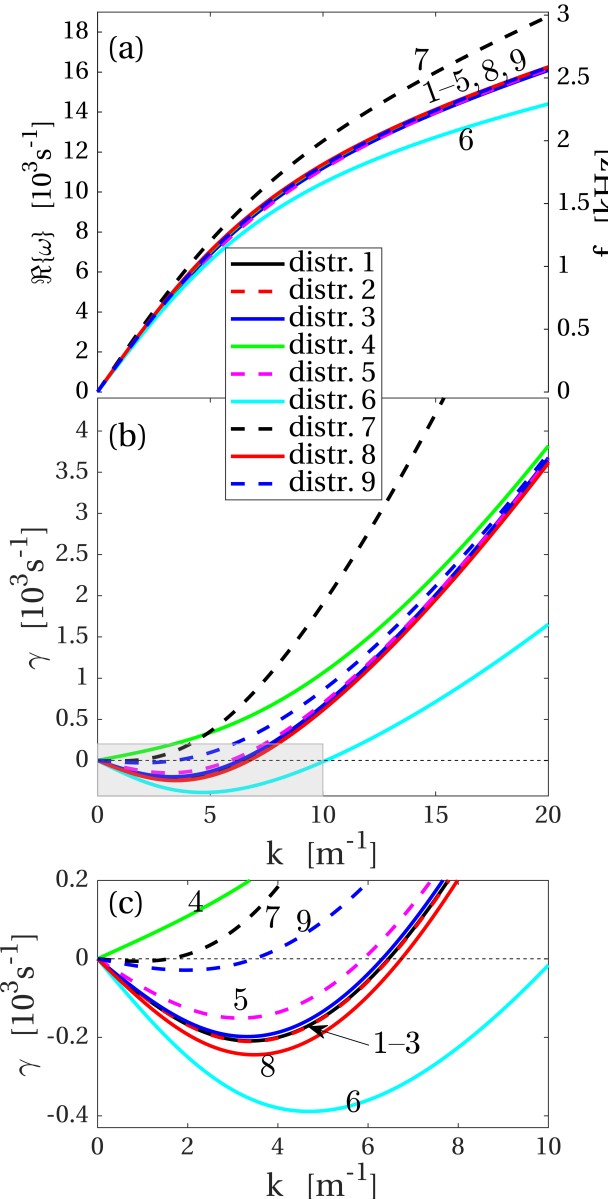

**Figure 6.** Dispersion relations at closest approach for the seven assumed distributions specified in Table 1. **(a)** real part of the dispersion relation. **(b)** damping rate $\gamma$. **(c)** zoom-in on the shaded rectangle in panel (b). The numbers next to the curves identify the different distribution functions shown in Table 1. Positive values of $\gamma$ correspond to wave damping and negative values to wave growth.

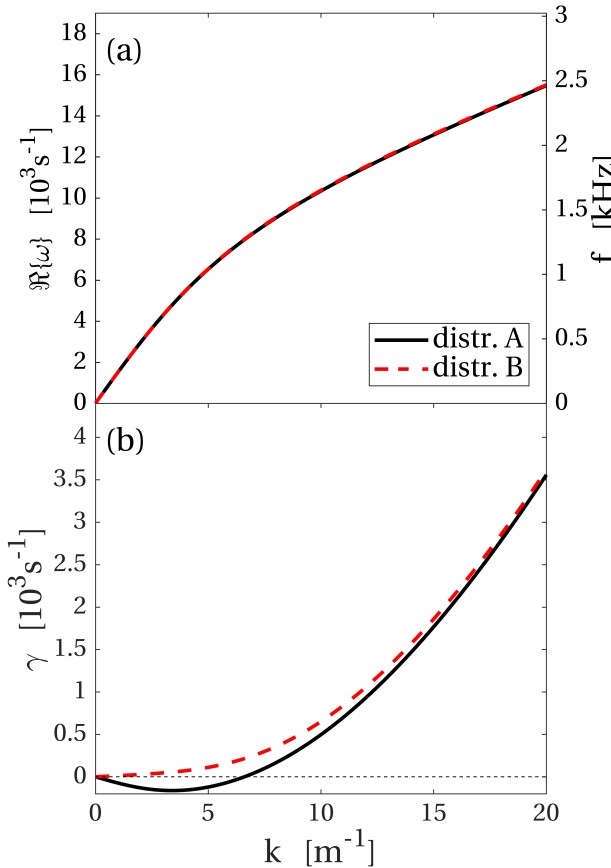

**Figure 7.** Dispersion relations while the spacecraft was outward bound based on assumed distributions specified in Table 1. **(a)** real part of the dispersion relation. **(b)** damping rate $\gamma$. Positive values of $\gamma$ correspond to wave damping and negative values to wave growth.

growth. The influence of suprathermal ions on the dispersion relations and growth rates is evaluated in Appendix B, and it is found to be similar to the difference between distribution with 0.01 and 0.02 eV ions. The distribution function of the cold ions cannot be measured directly, and hence effects caused by the shape of the distribution cannot be distinguished from effects caused by the temperature alone. However, we may conclude from all 9 cases that any process that gives the ions higher or the electrons lower energy will lead to decreased growth or increased damping.

Dispersion relations for distributions A and B, detailed in Table 1, are shown in Fig. 7. These distributions correspond to the plasma parameters obtained during the outbound passage of the spacecraft, and close to when the PSD represented by the blue line in Fig. 3 was recorded at 17:56:54. In distribution A the cold electrons have been given a drift velocity corresponding to the current density $J = 1.9\,\mu\mathrm{A\,m^{-2}}$ measured when the spacecraft was moving away, and the dispersion relation corresponding to distribution A is very similar to those at closest approach. In distribution B none of the populations have been assigned a drift

velocity. This leads to a stable distribution, and the waves are weakly damped instead of growing. Examining the magnetic

field in Fig. 2 we see no large scale change near 18:00. That the field remains constant, except for small fluctuations, while the spacecraft moves means that there is no large scale magnetic field gradient in this region and hence no large scale current either. Thus, the change in the plasma that affects the waves is the absence of a current, and this indicates that the reason why the wave spectrum fades out as the spacecraft moves away from the nucleus is the decline of the current density. Around 18:00 the waves likely were propagating to the spacecraft from a source region closer to the nucleus, where the current density was still high enough to generate waves.

## 4  Doppler shift

The dispersion relations are computed in the ion frame of reference and the observations are, by necessity, performed in a spacecraft-fixed frame. A frequency $f_\mathrm{m}$ in the moving medium is Doppler shifted to frequency $f_\mathrm{sc}$ in the spacecraft frame according to

$$f_\mathrm{sc} = \frac{v_\mathrm{ph}(f_\mathrm{m}) + u\cos(\alpha)}{v_\mathrm{ph}(f_\mathrm{m})} f_\mathrm{m}, \tag{6}$$

where $v_\mathrm{ph}$ is the phase velocity given by the dispersion relation, $u$ is the speed of the moving medium, and $\alpha$ is the angle between the wave direction of propagation and the velocity $\mathbf{u}$. Fig. 8a shows the $H_2O^+$ ion plasma frequency and the frequency of maximum growth, Doppler shifted to the spacecraft frame according to Eq. (6) for the dispersion relations that correspond to distributions 2, 7 and 9, and with $u$ determined by Eq. (2). The frequencies are shown as functions of the angle $\alpha$. The angle is not known from observations, but by comparing the Doppler shifted frequencies to the observed spectrum we can assess what values of $\alpha$ would lead to an interpretation in which observations and theory are consistent. This is explained in what follows.

The dispersion relations show that the damping is considerable at the ion plasma frequency. This has also been seen in experiments with current-driven ion acoustic waves, where the power declines with frequency and reaches the noise floor at frequencies well below the ion plasma frequency (Kawai et al., 1978). For our near closest approach sample spectrum shown by the black curve in Fig. 3 this happens at approximately 5 kHz. The range of angles $\alpha$ consistent with the observed spectra can then be constrained to those for which the ion plasma frequency is mapped to frequencies above 5 kHz. If the waves follow the dispersion relation corresponding to distributions 2 or 9, (solid red curve in Fig. 8a) this means that $\alpha \lesssim 56°$ and in the case of distribution 7 (solid black curve in Fig. 8a) the angle is restricted to $\alpha \lesssim 49°$. We will round this off to $\alpha \lesssim 50°$.

For a particular dispersion relation to be in agreement with observations, there should be significant wave power at the Doppler shifted frequency of maximum growth. With this regard distribution 7 is in better agreement with observations than distribution 2, because the spectrum has fallen significantly at 2 kHz and the dashed red curve in Fig. 8a is above 2 kHz for most of the relevant angle range of $\alpha \lesssim 50°$ determined above. The dashed black curve is close to 1 kHz in this range, and it is in good agreement with the peak of the spectrum in Fig. 3. The dashed blue curve corresponding to the frequency of maximum growth for distribution 9, in which cold and warm electrons both carry current in equal measure, falls between the the other two. Of the different dispersion relations we have examined it is the one corresponding to distribution 7 that best fits the Rosetta data. However, several distributions can lead to similar growth rates at similar frequencies, and we cannot constrain

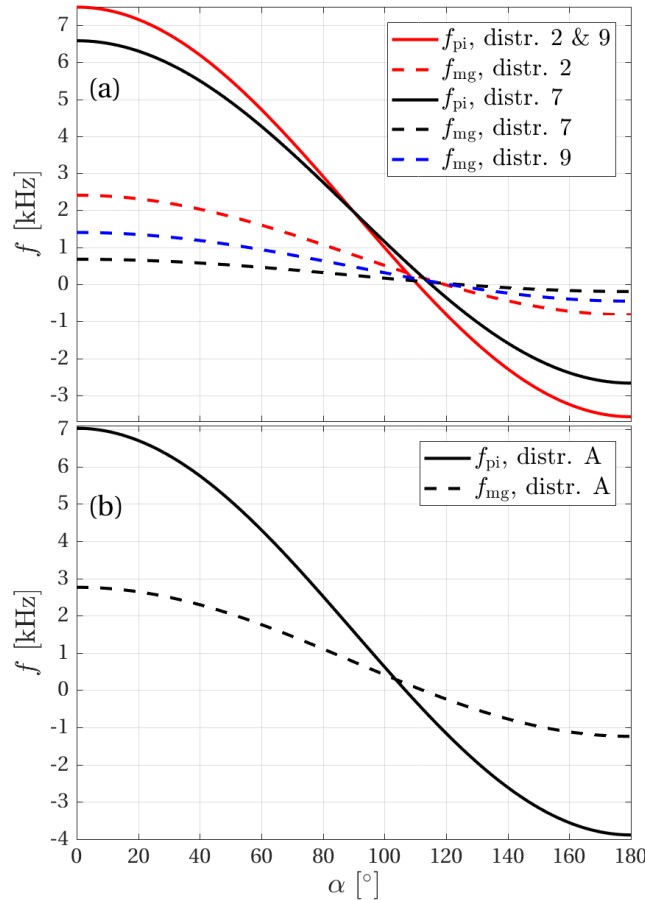

**Figure 8.** Doppler shifted $H_2O^+$ ion plasma frequencies, $f_{pi}$, and frequencies of maximum growth, $f_{mg}$, **(a)** at closest approach and **(b)** during the outbound motion of the spacecraft. The dispersion relations used to compute the Doppler shift correspond to distributions 2, 7, 9, and A in Table 1.

the distribution function closely. What we can say is that distributions that lead to moderate growth rates are in better agreement with the data than those that show very rapid growth.

For the outbound part of the spacecraft trajectory, Fig. 8b shows the Doppler shifted ion plasma frequency and frequency of maximum growth for distribution A. The real part of the dispersion relation for distributions A and B overlap in Fig. 7, and therefore the Doppler shifted ion plasma frequency will be the same for distribution B as for distribution A. For distribution B the waves are damped everywhere, and there is no frequency of maximum growth. We have already concluded in Sect. 3 that distribution B is more likely than distribution A, and that the waves that were observed as the spacecraft moved away were

not generated at the spacecraft location. The frequency where the wave power peaks tells us more about the source region than about the conditions at the spacecraft position. The blue curve in Fig. 3 has fallen to the noise floor at approximately $3\,kHz$, and from the solid curve in Fig. 8b the dispersion relation is seen to be in agreement with data for angles in the range $\alpha \lesssim 70°$.

## 5 Discussion and conclusions

We have analysed data obtained during Rosetta's close flyby of comet 67P on 28 March 2015. The multiple instruments used were RPC-LAP, RPC-ICA, RPC-MAG, and RPC-MIP, all part of the Rosetta Plasma Consortium. Waves which we interpret as current-driven ion acoustic waves were recorded by the Langmuir probe instrument RPC-LAP as probe current variations. These waves were seen all the time the spacecraft was close to the nucleus and the wave power started to decrease at approximately 24 km cometocentric distance. We estimated the current density from magnetic field measurements and found that the same currents that are involved in draping and pileup of the magnetic field (Koenders et al., 2016) are sufficient to drive the ion acoustic mode unstable, according to the kinetic model we have used to compute dispersion relations. Koenders et al. (2016) could observe field line draping until the rapid magnetic field change that occurred at 20:42 when Rosetta was at 34 km cometocentric distance.

Data from RPC-LAP indicate the presence of two electron populations, one cold at temperature around 0.2 eV and one warm at $\sim 4$ eV. Furthermore, the RPC-LAP characteristics show the presence of a cold ion population drifting with a speed between 3 km s$^{-1}$ and 3.7 km s$^{-1}$. Theoretical estimates of acceleration by a radial ambipolar electric field lead to ion bulk speeds in this range (Vigren and Eriksson, 2017) in agreement with observations (Odelstad et al., 2018). The cold component of the ion distribution went undetected by the ion spectrometer RPC-ICA. Instead a warm, several eV in temperature, ion distribution was detected by RPC-ICA, but its density is not sufficient for it to have any significant influence on the waves.

We are not able to measure the fine details of the electron and ion distributions. However, by testing different assumptions about the distributions it is possible to say something about it. We have seen that the best agreement between the theoretical dispersion relations and the observed wave spectra is obtained when the growth rate is moderate. This can be achieved with a cold ion distribution with $k_B T = 0.04$ eV. This is the warmest cold distribution that we tried, but it is still a very low temperature compared to all the other charged particle populations. The same result may be obtained with a lower temperature, if there also are suprathermal ions present. To accurately measure distribution functions at such low energies would represent a challenge in space-based instrumentation. These cold ion temperatures are reasonable, considering that Biver et al. (2019) found neutral temperatures up to approximately 0.02 eV between the nucleus and 15 km cometocentric distance. The ion distribution is formed by ionisation of the neutrals, and initially the neutral and ion temperatures are the same. On their way out to the spacecraft position, the ions may undergo some heating either through an increased bulk temperature or by forming suprathermal tails. The rapidly decreased growth with increasing cold ion temperature from 0.01 via 0.02 to 0.04 eV also shows that the cold ion temperature is of this order (a few hundredths eV). For a warmer cold ion population, the distribution would be stable and no waves generated, and as seen in Sect. 4, if the cold ion population is 0.01 eV or colder the higher growth rate leads to a worse agreement with observations. The growth rate is also influenced by what fraction of the current is carried by the cold electrons. This, in turn, depends on the relative speed of the two electron populations and how the electron density is distributed between them. To summarise the result of computing dispersion relations for the different distributions we have tried, it is distribution 7 in Table 1 that shows the best agreement with observations. It has the warmest cold ion distribution (0.04 eV), the current carried by the cold electrons, and no warm ion component as that was found to be negligible.

By computing the Doppler shift and comparing observed spectra with wave theory and known properties of current-driven ion acoustic waves we can estimate the angle between the bulk velocity of the cold ions and the propagation direction of the waves to be $\alpha \lesssim 50°$ for closest approach and $\alpha \lesssim 70°$ farther out when Rosetta was moving away and the wave power decreasing. Previous estimates have shown that ions move away from the centre of the comet, predominantly in a radial direction (Odelstad et al., 2018) as would be expected if they are accelerated by the ambipolar field present in the inner coma (Gunell et al., 2019). There are also observations of ions with an anti-sunward velocity component (Berčič et al., 2018), but those ions were faster than the $(3–3.7)\,\mathrm{km\,s^{-1}}$ we have observed here. If we assume that the ions move radially outward, the estimate of $\alpha \lesssim 50°$ for the waves near closest approach will also apply to the angle between the direction of propagation and the radial direction. Waves should propagate in the direction of the relative velocity between the electrons and the ions, and our angle estimates must not be seen as general results. They apply only at the position of the spacecraft during the flyby and for the orientation of the current at the time. During the outbound pass of the spacecraft, the angle of propagation cannot be restricted more than to say that it is below $70°$. Here, Rosetta was likely outside the source region, and the waves propagated to the spacecraft from a source located closer to the nucleus.

The use of wave observations in combination with wave theory complements the other measurements, particularly the cold ion population that is inaccessible to the particle instruments. It also lets us confirm the interpretation of probe data concerning the electron populations, and the interpretation of the observed variation of the magnetic field as a spatial gradient. Yet we only have information about the waves along a single spacecraft trajectory, and what we know about the current comes from crude estimates based on single spacecraft magnetic field observations. To obtain a more complete picture of currents and waves in the inner coma would require the comet to be accompanied by multiple spacecraft collecting data at the same time (Götz et al., 2019).

*Code and data availability.* The Rosetta data sets are available at the ESA Planetary Science Archive <https://archives.esac.esa.int/psa>. The specific data set used in this article is available at <https://doi.org/10.5281/zenodo.3973232> together with computer codes to produce the figures (Gunell et al., 2020).

**Appendix A: Dispersion relations**

In order to compute dispersion relations, the distributions described by Eq. (5) are written as a sum:

$$f(v) = \sum_j \frac{a_j}{v - b_j}, \tag{A1}$$

where $b_j$ are the poles of the distribution function and $a_j$ are the residues at those poles. The dielectric function for a plasma containing different species, $\alpha$, is (e.g. Krall and Trivelpiece, 1973)

$$\epsilon(k,\omega) = 1 + \sum_\alpha \frac{\omega_{p\alpha}^2}{k^2} \int \frac{k\,\mathrm{d}f_\alpha(u)/\mathrm{d}u}{\omega - ku}\,du. \tag{A2}$$

We normalise each population $f_\alpha$ and in Eq. (A2) it is weighted with its plasma frequency squared, $\omega_{\mathrm{p}\alpha}^2$. When integrating in the complex plane, the integral path is closed in the upper half plane, and $\epsilon(k,\omega)$ is expressed as (Gunell and Skiff, 2002)

$$\epsilon(k,\omega) = 1 - 2\pi i \sum_\alpha \omega_{\mathrm{p}\alpha}^2 \sum_{b_{j,\alpha} \in U} \frac{a_{j,\alpha}}{(\omega - k b_{j,\alpha})^2}, \tag{A3}$$

where $U$ denotes the upper half-plane. We assume a real value for $k$ and seek a complex $\omega$ by solving the equation for the dispersion relation, which is

$$\epsilon(k,\omega) = 0. \tag{A4}$$

This equation is solved numerically by seeking solutions that minimise $|\epsilon(k,\omega)|^2$. For more information about the method the reader is referred to the original articles (Löfgren and Gunell, 1997; Gunell and Skiff, 2001, 2002; Tjulin et al., 2000; Tjulin and André, 2002) and to the short review in a comet context by Gunell et al. (2017b).

## Appendix B: The influence of suprathermal ions

As mentioned in the main text the index $m$ controls the thickness of the suprathermal tails of the distribution function. In distributions 1–9, $m = 3$ for the ions and $m = 5$ for the electrons. For comparison we have performed calculations with $m = 6$ and $m = 8$ for all populations. These are shown in Fig. B1 as distributions 10 and 11 by the red and blue curve respectively. In both cases the cold ion temperature was 0.02 eV, and for comparison the dispersion relation for distribution 6, which has 0.01 eV cold ions, is also shown in Fig. B1. The results are very similar, and we conclude that the influence on the dispersion relation from the suprathermal tails is similar to the difference between distributions with 0.01 and 0.02 eV ions. Since we cannot directly measure the distribution function at these low energies we cannot tell the two effects apart. The distributions with higher $m$ indeces shown in Fig. B1 and those in Fig. 6 both agree with observations within the limits of experimental uncertainty.

*Author contributions.* HG performed the analysis in collaboration with CG, who also was the one to identify the flyby as an item of interest for wave studies, and EO, who in particular contributed to the plasma characterisation based on Langmuir probe data. All authors contributed to the writing of the final manuscript.

*Competing interests.* The authors declare that they have no competing interests.

*Acknowledgements.* This work was supported by the Swedish National Space Agency (grants 96/15 and 108/18). CG was supported by an ESA Research Fellowship. Work at LPC2E/CNRS was supported by CNES and by ANR under the financial agreement ANR-15-CE31-0009-01.

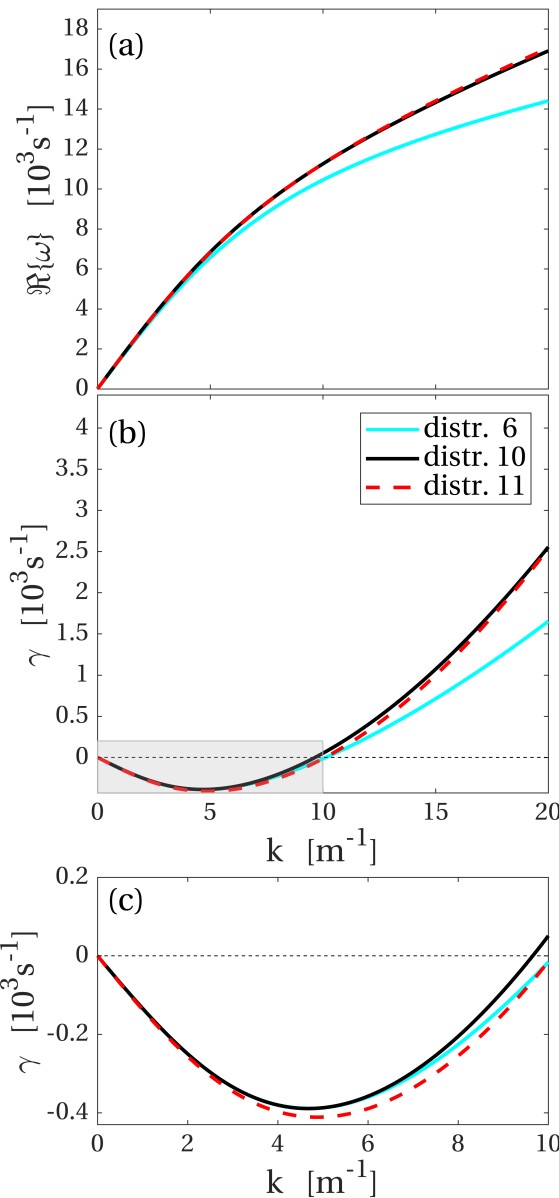

**Figure B1.** Dispersion relations for distributions with different suprathermal tails. Distribution 6 is the same as the distribution with that number shown in Fig. 6, and it has $m = 3$ for the ions and $m = 5$ for each of the two electron populations. Distribution 10 has $m = 6$ and distribution 11 $m = 8$ for all three populations. **(a)** real part of the dispersion relation. **(b)** damping rate $\gamma$. **(c)** zoom-in on the shaded rectangle in panel (b). Positive values of $\gamma$ correspond to wave damping and negative values to wave growth.

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
