# Peer review of "Ion acoustic waves near a comet nucleus: Rosetta observations at comet 67P/Churyumov-Gerasimenko"

_Annales Geophysicae, 2020_

## Referee Comment (RC1) · Martin Volwerk (Referee) · 24 Aug 2020

This paper discusses a fundamental plasmaphysical phenomenon: the ion acoustic wave, specifically the electric current-driven ion acoustic wave. Here, the current is defined as a velocity difference between the ions and the electrons in the ion rest frame. The authors use the RPC data from one close flyby of Rosetta by comet 67P/CG. The find the waves from just before closest approach to the comet and during egress up to a certain point. The "sudden" start is due to instrument operations the end is because of a quenching of the instability. The authors try to use the data from the different RPC instruments to characterize the plasma environment, but find that, because of

instrumental limitations (e.g. spacecraft charging) not all particle distribution functions can be obtained. After realistic assumptions on how to interpret the particle data, several different setups are made to calculate the dispersion relation of the ion acoustic waves. The authors show that one of the 7/2 scenarios works best to explain the observations.

This paper is well written, with the minor "complaint" that everything is well described in detail, but the authors do not spend any time of introducing the instability itself, although there are several citations to various papers (also by the lead author). It would be nice, for the interested reader, if the authors would add one short section on the specific instability.

However, I am not entirely sure how to interpret this paper, I have the feeling that there is more than just showing that the ion acoustic model can be described. Is it also the purpose of the authors to show that the observations of these waves can be a method to sample a parameter space of the plasma populations that cannot be measured by a charged spacecraft? If this is indeed the case (and it would be very useful) then this should be brought more forward, also in the abstract of the paper.

Some specific comments:

On line 169: "the wave frequency does not follow . . ." I would add here "of the observed waves" otherwise it reads a little confusing with the previous sentences talking about cyclotron waves.

On page 9 I am having trouble understanding the statement: "Cold ions are picked up by the electric field, moving along trajectories with a radius of curvature that is even larger." With the high density of the plasma that is measured by MIP, would this not mean that the spacecraft is in a pile-up region (B $\sim$ 30 nT) and thus the solar wind has been braked significantly. How much slow down do the authors expect? With 30 vs 5 nT, the "solar wind velocity" would probably be a factor 6 slower. What would the expected gyro radius be, then?

Line 225: "found to similar to the difference" here "be" is missing.

Line234: "Examining the magnetic field in Fig. 2 we see no large scale change near 18:00, which means that there was no large scale current present." I am not sure that is correct. The magnetic field remains the same, if suddenly a large scale current would disappear (the one driving the IA waves) then would one not likely see a change in the magnetic field, like just before 21:00?

Line 266: Here Fig. 8b is discussed, and it would be good if the authors would put in that that is for the outbound part of the orbit.

---

## Referee Comment (RC2) · Anonymous Referee #2 · 27 Oct 2020

Ion acoustic waves near a comet nucleus : Rosetta observations at comet 67P/Churymov-Gerasimenko, by Gunell et al.

**General comments:**

This paper addresses observations of ion acoustic waves (IAW) in the vicinity of 67P nucleus a few months before perihelion. The paper is based on observations of 4 instruments of the Rosetta Plasma Consortium and of ROSINA-COPS. The IAW are observed in the region of high current drift (near closest approach to the nucleus) in connection with high current drift, while they were not further from the nucleus where there was no significant current drift observed.

The paper is in general written in good style, but the formulation of physical processes is sometimes more qualitative than quantitative, even vague at times. The identification of the IAW lacks clear characterization of their properties. Some numbers of the observed or derived parameters are given but not always substantiated.

Reference is made to the first detection paper of IAW at 67P. The detection conditions, including the plasma parameters seem to be different, but those differences, and their consequences on the IAW properties may not discussed in depth.

Beyond the plasma physics interest of the detection of IAW in 67P environment, In the discussion, I would have like to see a short paragraph discussing the interest of the detection of IAW in terms of cometary physics and Cometary sciences. As written in the first sentence in the introduction, "observations of waves can give us information of the plasma physics in which they are generated and through which they have travelled" (a rather strange formulation by the way). In the discussion I would have expected reading something of what the detection and characterization of IAW bring in terms of understanding the comet plasma (and neutral ?) environment how do they help in constraining physical processes at work inside the coma.

The abstract may not fully reflect the content of the paper.
It should include something on:
- Hot ions are not contributing the IAW
- IAW waves are detected when a current flow is present as determined from B-field measurements
- The high spacecraft charging complicates the interpretation of the observations.

**Detailed comments are provided below.**

Line 10. Replace « travelled » by « propagated » ?

Line 12: Add "charged" in front of "particle"

It's hard to appreciate the importance of the Doppler shift without the mentioning of the frequency range of the waves. Explain the relation between the bulk velocity and the Doppler shift.

Line 21: Provide the parameters that lead to a LHF < 15 Hz.

Line 23-26: Is there a relation between the steepened waves only observed outside the diamagnetic cavity and the waves in the LHF range observed on both sides

At this stage, it would be useful to recall how IAW modes are identified.
During the review, I just came acrosss recently published paper (that was not available at the time of the paper submission) that addresses well the identification of IAW modes (Mozer et al. 2020). One characteristic used is the phase. Can information about the phase be obtained with the LAP probe signal ?. Would the availability of both P1 and P2 signals (although with different amplitudes) help in obtaining information about the phase?

Line 28: Define undefined variables, e.g. omega, k, kB, mi. The need to define variables used applies also to other parts of the manuscript.

Line 33-40. It is said that no diamagnetic cavity was seen during the flyby studied,. The plasma conditions, and characteristics of the IAW confined in the diamagnetic cavity (Gunell et al, 2017) seem to be somewhat different from those reported in this article (no diamagnetic cavity had formed). There is no discussion of the similarities and differences between the two studies.

Line 41: "likely" is a vague statement.  Could it be that the cavity had formed closer to the nucleus, where the s/c did not go on that flyby ?.

Line 43:  explain the implication of the infant bow shock (from simulation) and the fact that the diamagnetic cavity had not formed (or was not observed).

Line 46: Not sure CSEQ is known to all potential readers. A reference, beyond the definition that follows, would be desirable.

Line 56: replace "plasma waves" by "probe current variations attributed to waves"

Line 57: clarify how the probe current variation relates to plasma waves

Line 59: You may not have said before that there were two Langmuir Probes

Line 61: It would be useful to indicate the value of the S/C potential, in order to better appreciate the difference between a probe at +30 V and one at -30V.

Line 62: Was the bulk speed of the ions "measured" or "estimated"?. See further questions later

Line 67: add "magnetic field" in "The (magnetic field) components"

Figure 2:

Would be useful to say in the legend or to write in the top two panels the bias value for each of the two LAP probes.

Is there a physical explanation for the sharp transition of the plasma density derived from MIP measurements before and after 10:00 while the RPC-ICA spectra are quite similar.
Any explanation as to why there are no RPC-MIP measurements after 20:00?

Legend: I would say the plasma density is "derived" rather than "measured" from the RPC-MIP measurements.

Line 73-74: Not obvious in the figure that the frequency scale starts at 200 Hz. Setting the origin of the Y-scale at 0 and leaving the space between 0 and 200 Hz blank would make it clear.

Line 75: Not clear if signal is a wave signal or noise enhancement. It would be useful to show the non-Doppler shifted LHF line for reference

Line 77-78. I would put it the other way around. Probe 1 being dominated by electron current, and probe 2 being dominated by ion current, leads to the fact that the power spectral density of probe 1 is several orders of magnitude higher.

Line 77: clarify « ..... signal proportional to the density variation of the waves » should be substantiated

Explain why the wave signal observed is identified as IAW. What are the wave characteristics that allow to infer that?

Line 80: This statement about the plasma density being comparable in both events is not verifiable as the density measurements are not illustrated in Fig. 3 of the 2017 paper.

Line 82: Clarify "condition when signal is observed trough displacement current (capacitive coupling) vs particle current? If I understand well, a decrease of a factor of 10 of the current implies that the wave can no more be detected by the current variation, but instead by capacity coupling. This statement should be elaborated.

Line 97: Provide the reference to the publication for the artefacts ? It seems that the artefacts are harmonics of 1 kHz as well evidenced in the PDS line at 19:08:54. This is within the range of the Fce value given in line 168. Can it be excluded that those "artefacts" are harmonics of Fce. It would be desirable to provide the value of Fce for the period. Can it be ruled out that part of the detected noise is enhanced (excited) by those artefacts ? Are those artefacts discernable in the P2 data?

Line 101-102: How do you quantify plasma inhomogeneities at 10%. What is the accuracy of the MIP measurements?

Line 104-105: Process by which the ions are getting heated (6eV) and how this temperature is derived?

Line 105: You should say at least once that those are positive ions. Apologies if it was said before and I missed it.

Are there also negative ions present in the plasma? if yes, how would those negative ions affect the IAW generation and damping ?

Line 107. I suppose the fit is performed with a drifting maxwellian population. Please confirm

Line 110-115. I Not convincing argument as to why most of the ion population is not visible, all ions (warm and cold) should be accelerated by the S/C potential, should they not ?

I have difficulties to follow the reasoning about the non-detectability of the cold ions. Should they not be accelerated to 20 V as well. If the fraction of that population that is detected may not be distinguishable from the ions belonging to the warm population, should they not appear in the maxwellian fit described earlier. This seems to be somewhat in contradiction when saying that it may explain that the cold water ion population (still accelerated to 20 V) is invisible to RPC-ICA. "May" means that there could be other explanations. Please elaborate.

Line 116: Clarify how the various photoemission current is taken into account in the ion part of the I-V curve.

Line 116: Discuss the deviation from linearity of the ion portion of the I-V curve at negative potential clearly visible at 13:25:26, but also discernable at 17:52:06. It is said earlier that the I-V curve is acquired in the -30 +30 V range. If so, it would be interesting to show the hidden part of the curve, between -30 V and -30 V.

My examination of the I-V curve indicates that the local plasma potential is about 20 V, confirming that the S/C is charged to about -20V. The energy of the ions hitting the probe may reach 50 V (60 V if the probe is polarized at -30 V). In this energy range, is it possible that secondary emission plays a role? is photoemission of the probe surface taken into account in the probe current ?

In eq (1), define variable V. Is such a formula directly applicable for a drifting ion population?. The hot ion population does not seem to be considered in the overall ion current. Justify. Discuss the applicability of eq (1) to the current plasma conditions

Taking the ion density equal to the plasma density ignores the hot ion population contribution. Is this justified?

A formulation of the I-V curve taking into account all current contributions should be written.

A proper discussion of the various measurement uncertainties would be desirable

Probably not surprising that the numbers are within the range of those observed by Odelstad et al. (2018) if the same method of analysis is used (I did not check that point). Point to be clarified.

Line 126: Replace " a upper " with "an upper"

Line 129: Not clear why the Biver et al 0.02 eV neutral temperature is compared to the 1eV (ion) kinetic energy. Please elaborate the argument.

Line 136. The slope of the two electron current curves are clearly different. Why do they provide the same value of Te (about 0.2 eV)?

Not clear how the plasma potential is estimated to 12 and 14 volts. Elaborate. My estimation is more around 20 V (see above). In fact, the plasma potential is derived from a measurement made inside the plasma sheath of the charged spacecraft. Discuss the uncertainty of this value?

When revising the paper, I would advice to discuss this spacecraft charging effects with reference to the recently published paper by Johansson et al. https://doi.org/10.1051/0004-6361/202038592

Discuss uncertainties in the derived numbers

Line 143: Confirm that, in the presence of two equal-density electron populations, the MIP max represents the plasma frequency (how is it defined with two such different populations). It is noted that the MIP phase data are not referred to. Are they consistent with the amplitude data?

Line 162-163: What is the implication of the measurement uncertainty expressed by the sentence "Thus, the current density may have been both higher and lower that these average values during the flyby" ?

Line 169: How is the wave frequency characterized? justify the important affirmation that the wave frequency does not follow the change of the magnetic field, used to justify that the waves observed are not electron cyclotron waves.  Provide values.

Line 172: How was the "typical length" for the variation in wave amplitude deduced to 10 km ?. What is meant by "typical length"

Line 177: clarify if you refer to electron or ion gyro-radius, or both

Table 1: Clarify parameters used. Electron VD?

Line 184: the uncertainties in the measurements is not well reflected in the values reported in sect; 2. Can the measurement uncertainties be quantified?

Line 190 -192
It would be desirable to provide the formula of the dispersion relation used, although indeed, proper reference is given. May be as important, if not more, than the formula for the distribution function.

Line 194: define variable v? is variable "vd" used in the formula the same as "vD" used in the table ? use consistent notation.

Line 211: The non-effect of the warm ions (the one detected by ICA) lead to consider the cold ions whose density is set equal to the « measured » electron density. This makes a strong assumption that the plasma is locally neutral, which may not be the case in the sheath around the spacecraft. Justify.

Line 214: is it justified to assign a drift velocity to only one of the two electron populations?

Line 216: Such a strong conclusion should be more substantiated.

Line 225: word missing: « ...is found to (be) similar.. »

Line 245-248, and legend Fig 8: The notations used should be all defined (in the legend)

Line 250-251: Not clear what means a « reasonable spectrum » and how this observation is reached.

Line 251-273: The narrative discussions seem to be very qualitative. Not clear that the conclusions reached are well substantiated.

Line 275: specify that a multi-instrument data set was analyzed. Recall which data set were used.

Line 276: indicate that the waves were recorded as Langmuir probe current variations

Line 283-284: The ion drift value, obtained from the analysis of the LAP I-V curve, was questioned above. What is the process causing the ion drift speed of 3 to 3.7 km/s. Could this be partly a local phenomena inside the sheath of the charged spacecraft?

Line 285: replace "electron volt" by "eV"

Line 288: « ... possible to say something about it ». I found this statement very speculative with the limited cases tested.

Line 295: remove « the » before « bulk »

Line 297: Discuss the processes that would increase the bulk temperature or form supra-thermal tails. Can wave-particle interaction contribute to the ion heating process ?

Current carried by cold electrons?

Line 313-316: Indeed, it seems pretty strong conclusions are reached from crude estimates and measurements with uncertainties (which are not quantified).

---

## Author Comment (AC1) · 24 Nov 2020

We respond below to the comments raised by the reviewer, setting his words in *italics* with our response following below.

**Response to Referee 1 (Volwerk)**

*This paper discusses a fundamental plasmaphysical phenomenon: the ion acoustic wave, specifically the electric current-driven ion acoustic wave. Here, the current is defined as a velocity difference between the ions and the electrons in the ion rest frame. The authors use the RPC data from one close flyby of Rosetta by comet 67P/CG. The find the waves from just before closest approach to the comet and during egress up to a certain point. The "sudden" start is due to instrument operations the end is because of a quenching of the instability. The authors try to use the data from the different RPC instruments to characterize the plasma environment, but find that, because of instrumental limitations (e.g. spacecraft charging) not all particle distribution functions can be obtained. After realistic assumptions on how to interpret the particle data, several different setups are made to calculate the dispersion relation of the ion acoustic waves. The authors show that one of the 7/2 scenarios works best to explain the observations.*

*This paper is well written, with the minor "complaint" that everything is well described in detail, but the authors do not spend any time of introducing the instability itself, although there are several citations to various papers (also by the lead author). It would be nice, for the interested reader, if the authors would add one short section on the specific instability.*

*However, I am not entirely sure how to interpret this paper, I have the feeling that there is more than just showing that the ion acoustic model can be described. Is it also the purpose of the authors to show that the observations of these waves can be a method to sample a parameter space of the plasma populations that cannot be measured by a charged spacecraft? If this is indeed the case (and it would be very useful) then this should be brought more forward, also in the abstract of the paper.*

We have included more discussion on how the combination of wave observations and theory supplements the other measurements in the discussion section and also in the abstract. We have put some information on the current-driven ion acoustic instability in the introduction – next to where ion acoustic waves are mentioned.

*On line 169: "the wave frequency does not follow . . ." I would add here "of the observed waves" otherwise it reads a little confusing with the previous sentences talking about cyclotron waves.*

Changed as suggested. We also inserted a sentence giving specific numbers in response to the other referee.

*On page 9 I am having trouble understanding the statement: "Cold ions are picked up by the electric field, moving along trajectories with a radius of curvature that is even larger." With the high density of the plasma that is measured*

*by MIP, would this not mean that the spacecraft is in a pile-up region (B ∼ 30 nT) and thus the solar wind has been braked significantly. How much slow down do the authors expect? With 30 vs 5 nT, the "solar wind velocity" would probably be a factor 6 slower. What would the expected gyro radius be, then?*

Assuming an initial speed of 400 km/s, slowed down by a factor of 6, the gyro radius of an $H_2O^+$ ion in a 30 nT magnetic field would be 415 km, which is "even larger" than the 50 km found from the warm ion thermal speed.

*Line 225: "found to similar to the difference" here "be" is missing.*

Corrected.

*Line234: "Examining the magnetic field in Fig. 2 we see no large scale change near 18:00, which means that there was no large scale current present." I am not sure that is correct. The magnetic field remains the same, if suddenly a large scale current would disappear (the one driving the IA waves) then would one not likely see a change in the magnetic field, like just before 21:00?*

It seems this statement can be misunderstood. A change, as for example the disappearance of a large scale current, would indeed be seen in the magnetic field. We have replaced the sentence by the following (the first half of the original sentence still remains), which we think should be clearer:

"That the field remains constant, except for small fluctuations, while the spacecraft moves means that there is no large scale magnetic field gradient in this region and hence no large scale current either. "

*Line 266: Here Fig. 8b is discussed, and it would be good if the authors would put in that that is for the outbound part of the orbit.*

We have inserted the following clause at the beginning of the first sentence of the paragraph:

"For the outbound part of the spacecraft trajectory,"

---

## Author Comment (AC2) · 24 Nov 2020

We respond below to the comments raised by the reviewer, setting the reviewer's words in *italics* with our response following below.

**Response to Referee 2**

This paper addresses observations of ion acoustic waves (IAW) in the vicinity of 67P nucleus a few months before perihelion. The paper is based on observations of 4 instruments of the Rosetta Plasma Consortium and of ROSINA-COPS. The IAW are observed in the region of high current drift (near closest approach to the nucleus) in connection with high current drift, while they were not further from the nucleus where there was no significant current drift observed.

The paper is in general written in good style, but the formulation of physical processes is sometimes more qualitative than quantitative, even vague at times. The identification of the IAW lacks clear characterization of their properties. Some numbers of the observed or derived parameters are given but not always substantiated.

Reference is made to the first detection paper of IAW at 67P. The detection conditions, including the plasma parameters seem to be different, but those differences, and their consequences on the IAW properties may not discussed in depth.

We have amended the manuscript, and we think that it now adequately describes how the ion acoustic waves were identified, that the derived parameters are sufficiently substantiated.

Beyond the plasma physics interest of the detection of IAW in 67P environment, In the discussion, I would have like to see a short paragraph discussing the interest of the detection of IAW in terms of cometary physics and Cometary sciences. As written in the first sentence in the introduction, "observations of waves can give us information of the plasma physics in which they are generated and through which they have travelled" (a rather strange formulation by the way). In the discussion I would have expected reading something of what the detection and characterization of IAW bring in terms of understanding the comet plasma (and neutral ?) environment how do they help in constraining physical processes at work inside the coma.

The abstract may not fully reflect the content of the paper. It should include something on:

- Hot ions are not contributing the IAW
- IAW waves are detected when a current flow is present as determined from B-field measurements
- The high spacecraft charging complicates the interpretation of the observations.

We have put all those things in the abstract, and we have also added text to the discussion section to highlight how wave observations in combination with wave theory supplement the measurements by the other instruments and the interpretation thereof.

Line 10. Replace « travelled » by « propagated » ? Changed as suggested.

*Line 12: Add "charged" in front of "particle"* Changed as suggested.

> It's hard to appreciate the importance of the Doppler shift without the mentioning of the frequency range of the waves. Explain the relation between the bulk velocity and the Doppler shift.

This must be referring to the mentioning of the Doppler shift in the abstract. We have modified the sentence in questions so that it also states the frequency range of the waves in the plasma frame.

"Near closest approach the propagation direction was within 50° from the direction of the bulk velocity, leading to a Doppler shift of the waves, which in the plasma frame appear below the ion plasma frequecy  $f_{\rm pi} \approx 2 \,\mathrm{kHz}$ , to the spacecraft frame where they cover a frequency range up to approximately  $4 \,\mathrm{kHz}$ ."

**Line 21: Provide the parameters that lead to a LHF**

Figure A: Time series of the probe currents for two times during the day of 28 March 2015.

facts is significantly below that of the wave signal until that signals starts to fade away. The artefact may excite a low amplitude wave at the isolated frequencies where they are seen, but not the much higher amplitude waves of interest here, which also are present in a much wider frequency range than the artefacts. See also the figure next to this piece of text (Fig. A) for two time series of the probe current.

Line 101-102: How do you quantify plasma inhomogeneities at 10%. What is the accuracy of the MIP measurements?

See the electron density plot to the right. The accuracy of the data in the figure is  $5-10 \text{ cm}^{-3}$ , the variations are one order of magnitude larger, and the mean is lager by yet another order of magnitude. The accuracy of the MIP measurements is also about 10%. The accuracy of the electron density from the MIP measurement is com-

**Figure B:** *LAP* plasma density sample figure for this day.

puted from the frequency resolution of the instrument, which measures the plasma frequency from the identification of the plasma frequency line in the mutual impedance spectra.

**Line 104-105: Process by which the ions are getting heated (6eV) and how this temperature is derived?**

We merely observe those ions. The heating process is outside the scope of this paper. The temperature is obtained from the same fit that gave the density, mentioned a few lines below. We have introduced words to say that the temperature is also obtained by fitting.

**Line 105: You should say at least once that those are positive ions. Apologies if it was said before and I missed it.**

We inserted the word positive before ions in the first sentence of this paragraph. In other places in the manuscript we assume the ions to be  $H_2O^+$  ions.

Are there also negative ions present in the plasma? if yes, how would those negative ions affect the IAW generation and damping ?

There are no significant amounts of negative ions in the plasma, at least not from a plasma physics point of view. Measurements at comet 1P showed that fewer than one ion in  $10^4$  are negative.

Line 107. I suppose the fit is performed with a drifting maxwellian population. Please confirm

Confirmed. We updated the text to say "drifting Maxwellian".

Line 110-115. I Not convincing argument as to why most of the ion population is not visible, all ions (warm and cold) should be accelerated by the S/Cpotential, should they not ?

I have difficulties to follow the reasoning about the non-detectability of the cold ions. Should they not be accelerated to 20 V as well. If the fraction of that population that is detected may not be distinguishable from the ions belonging to the warm population, should they not appear in the maxwellian fit described earlier. This seems to be somewhat in contradiction when saying that it may explain that the cold water ion population (still accelerated to 20 V) is invisible to RPC-ICA. "May" means that there could be other explanations. Please elaborate. They are accelerated by the spacecraft potential, but the ability of the instrument to detect low energy ions is severely limited. The sensitivity of the detector is low for such low energies, and the angular range that allows entry into the instrument narrow. Thus, a mono-energetic low energy beam, which the cold ion population is, can go completely undetected. We described that in terms of a field of view, vanishing at low energies. This description seems to be inefficient as a form of communication, and we have changed the text to provide a better explanation.

"The sensitivity of RPC-ICA is low in the lowest energy range, and the angular range that allows entry into the instrument is narrow. A mono-energetic low energy beam is liable to be undetected. Thus, the discrepancy between the RPC-ICA measured ion density and the plasma density measured by RPC-MIP is explained by a cold water ion distribution that is invisible to RPC-ICA."

Finally, to avoid misunderstanding, of course the Maxwellian fit is to the part of the distribution that is above the limit for detection.

**Line 116: Clarify how the various photoemission current is taken into account in the ion part of the I-V curve.**

The photoemission comes in as an offset in the ion current (below the plasma potential), and as such, does not play a role in the slope of the ion current. Also, LAP1 is in shadow behind the spacecraft for the first sweep. We added the following at the end of the paragraph:

"Photo emission comes in as an offset in the ion current, below the plasma potential, and as such, does not play a role in the slope of the ion current. Furthermore, for the first of the sweeps, the probe was in shadow behind the spacecraft."

Line 116: Discuss the deviation from linearity of the ion portion of the I-V curve at negative potential clearly visible at 13:25:26, but also discernable at 17:52:06. It is said earlier that the I-V curve is acquired in the -30 + 30 V range. If so, it would be interesting to show the hidden part of the curve, between -30 V and -30 V.

It is possible that the deviation from linearity is the onset of secondary emission from ion impact. However, it is not always present, does not behave very similarly from sweep to sweep, is sometimes nonlinear and wavelike in nature, As a full Langmuir probe sweep takes 3.2 seconds, (i.e. is not instantaneous), it is more likely that this is a capacative coupling to some excited wave propagated by ions with an observed frequency near 1 Hz and grows larger in amplitude when more ions are attracted by the probe. This is also confirmed to be present when looking at the continuous (60 Hz) ion current

**Figure C:** *LAP* characteristics starting from the beginning of the sweep.

data from LAP 2. Therefore, we restrict the ion current fit to a region below the spacecraft potential where the ion current is linear, and any wave amplitude is small. A figure showing the "hidden" part is shown here (Fig. C). The figure in the paper has also been updated but in a different way with some new panels.

My examination of the I-V curve indicates that the local plasma potential is about 20 V, confirming that the S/C is charged to about -20V. The energy of the ions hitting the probe may reach 50 V (60 V if the probe is polarized at -30 V). In this energy range, is it possible that secondary emission plays a role? is photoemission of the probe surface taken into account in the probe current ?

We have reexamined the I-V curve and agree with the local plasma potential and spacecraft potential estimation.

Regarding secondary emission, it is certainly possible, but the quantum yield of ion impact ionisation is low for metals particularly at low energies, less then unity. And for a ceramic surface layer like TiN, it is likely that the quantum yield is even lower. We have added a sentence explaining that if secondary emission is present also at the low (10eV) energy range where our ion current analysis begins, then this would inflate our ion velocities with a factor equal 1+ the quantum yield. It would only have a very limited impact on Probe current variations attributed to waves.

Regarding photoemission, the LAP1 is in shadow between 11 and 14:30 and is not photoemitting. Also, at the potentials the ion current analysis is performed, the photoemission is an offset, and does not play a role in determining the effective ion speed.

In eq (1), define variable V. Is such a formula directly applicable for a drifting ion population?. The hot ion population does not seem to be considered in the overall ion current. Justify. Discuss the applicability of eq (1) to the current plasma conditions

The variable V, representing the probe to plasma potential, was already defined above the equation together with the bulk speed u and the current I. The warm ion population is neglected because of its low density, and the equation is applicable because the thermal speed of the cold ions is far below their bulk speed. A sentence to that effect has been introduced just before Eq. (1).

**Taking the ion density equal to the plasma density ignores the hot ion population contribution. Is this justified?**

Yes, it is. The warm ion density is  $4 \text{ cm}^{-3}$  and the plasma density  $1600 \text{ cm}^{-3}$ . The factor of 400 between them ensures that the error is negligible.

**A formulation of the I-V curve taking into account all current contributions should be written.**

With the changes made outlined above, we feel that all relevant current contributions have been described.

**A proper discussion of the various measurement uncertainties would be desirable**

We answer this now and refer back here, whenever this comment reappears below. We have written a new section about the uncertainties. We have not removed any existing

discussions from the other section, which leads to some overlap, but we don't find that overly troubling. The new section appears at Sect. 2.5 in the revised manuscript.

Probably not surprising that the numbers are within the range of those observed by Odelstad et al. (2018) if the same method of analysis is used (I did not check that point). Point to be clarified.

We do not claim it to be surprising. However, since the observations by Odelstad et al. (2018) were performed in a different magnetic field environment and at a different distance to the nucleus, it was no certainty beforehand that the results would be similar. We have added text, clarifying the different conditions.

*Line 126: Replace " a upper " with "an upper"* The sentence has been removed in its entirety as, under careful examination, it has been found to be incorrect.

Line 129: Not clear why the Biver et al 0.02 eV neutral temperature is compared to the 1eV (ion) kinetic energy. Please elaborate the argument.

We added the sentence

"This confirms the assumptions stated above Eq. (1) and ensures the applicability of the equation."

Line 136. The slope of the two electron current curves are clearly different. Why do they provide the same value of Te (about 0.2 eV)?

As the slope is proportional to both electron density and electron temperature, two different slopes measured at plasmas with different densities can have the same temperature. Also, for the temperatures obtained here, the values 0.16 and 0.22 were both rounded off to 0.2.

Not clear how the plasma potential is estimated to 12 and 14 volts. Elaborate. My estimation is more around 20 V (see above). In fact, the plasma potential is derived from a measurement made inside the plasma sheath of the charged spacecraft. Discuss the uncertainty of this value?

We have made a new assessment of the plasma potential, and also of the warm electron temperature. The new version of the manuscript says that the plasma potential is approximately 20 V and the warm electron temperature  $k_{\rm B}T_{\rm ew} = 2 \,{\rm eV}$ .

When revising the paper, I would advice to discuss this spacecraft charging effects with reference to the recently published paper by Johansson et al. https://doi.org/10.1051/0004-6361/202038592

We have added the following sentence in connection with the determination of the spacecraft potential.

"It was shown in simulations by Johansson et al. (2020) that the spacecraft potential is driven negative by positively biased elements on the solar panels that collect cold electrons from the plasma." Discuss uncertainties in the derived numbers

See our response to this comment on page 9.

Line 143: Confirm that, in the presence of two equal-density electron populations, the MIP max represents the plasma frequency (how is it defined with two such different populations). It is noted that the MIP phase data are not referred to. Are they consistent with the amplitude data?

In the presence of two electron populations (for instance two equal-density electron populations), the main resonance in the mutual impedance spectra is indeed associated to the (total) electron density, and give therefore access to the total electron density. In some cases (e.g. large temperature ratio and similar densities such as for two equal-density electron populations) a second resonance associated to electron acoustic waves generated in the plasma by the MIP electric transmitter. The general instrumental response of the MIP mutual impedance probe in a plasma characterised by two electron populations with different temperatures is described in Gilet et al. 2017 <a href="https://doi.org/10.1002/2017RS006294">https://doi.org/10.1002/2017RS006294</a>> and Wattieaux et al, 2020

<https://doi.org/10.1051/0004-6361/202037571>. We have added the following sentence to the manuscript:

"The general instrumental response of the RPC-MIP mutual impedance probe in a plasma characterised by two electron populations with different temperatures is described by Gilet et al. (2017) and Wattieaux et al. (2020)."

Line 162-163: What is the implication of the measurement uncertainty expressed by the sentence "Thus, the current density may have been both higher and lower that these average values during the flyby"?

See our response to this comment on page 9.

Line 169: How is the wave frequency characterized? justify the important affirmation that the wave frequency does not follow the change of the magnetic field, used to justify that the waves observed are not electron cyclotron waves. Provide values.

We inserted the following sentence to provide values:

"For example, the PSD peaks at 700 Hz for both times 13:24:54 and 15:16:54, shown by the black and red curves in Fig. 3, respectively, even though the electron cyclotron frequency was 1.1 kHz at 13:24:54 and 0.57 kHz at 15:16:54."

Line 172: How was the "typical length" for the variation in wave amplitude deduced to 10 km?. What is meant by "typical length"

This is described in the sentence immediately before the one where the word "typical" appears:

"The spacecraft was at 15 km cometocentric distance at closest approach and at 25 km at 18:00 when the wave amplitude started to decrease. Thus, the typical length for the variation in wave amplitude is about 10 km."

Line 177: clarify if you refer to electron or ion gyro-radius, or both

We have replaced "particles" by "electrons and ions".

Table 1: Clarify parameters used. Electron VD?We added an explanation of the notation to the table caption.

Line 184: the uncertainties in the measurements is not well reflected in the values reported in sect; 2. Can the measurement uncertainties be quantified? See our response to this comment on page 9.

Line 190 -192 It would be desirable to provide the formula of the dispersion relation used, although indeed, proper reference is given. May be as important, if not more, than the formula for the distribution function.

We have updated the text to provide that, but since this is somewhat involved and requires new notation to be introduced and several equations to be displayed, we have put it in a new appendix, which is now Appendix A.

Line 194: define variable v? is variable "vd" used in the formula the same as "vD" used in the table ? use consistent notation.

v means velocity. This piece of information has now been added to the manuscript. We have also changed all instances of  $v_{\rm d} v_{\rm D}$ .

Line 211: The non-effect of the warm ions (the one detected by ICA) lead to consider the cold ions whose density is set equal to the « measured » electron density. This makes a strong assumption that the plasma is locally neutral, which may not be the case in the sheath around the spacecraft. Justify.

That the warm ions, due to their low density, have no influence on the wave properties does not tell us anything about the neutrality of the plasma. Also, the probe is outside the sheath as the Debye length, as estimated below, is only 26 cm.

Line 214: is it justified to assign a drift velocity to only one of the two electron populations?

As collisions cools the electrons to give them different temperatures they cal also obtain different velocities. However, this comment inspired another test, namely assigning equal drift velocities to both electron population. This is now distribution 9. The result is reported in the revised manuscript:

"In distribution 9 the current is carried by both electron populations, which each are given a drift velocity of  $|\mathbf{v}_{\rm D}| = 19.6 \,\rm km \, s^{-1}$ . This yields a lower growth rate than for distribution 2, but the mode is still unstable for a range of k values."

Distribution 9 is now also included in Fig. 8 and the discussion thereof (see the revised manuscript).

Line 216: Such a strong conclusion should be more substantiated.

Following the evaluation of distribution 9 this statement has been moderated:

"We conclude that to drive the ion acoustic waves unstable the current cannot be carried by the warm electrons alone."

This is substatiated by the computations based on distribution 9.

Line 225: word missing: « ... is found to (be) similar.. » Corrected. Thanks also to referee 1.

Line 245-248, and legend Fig 8: The notations used should be all defined (in the legend)

Notation explanation may now be found in the caption.

Line 250-251: Not clear what means a « reasonable spectrum » and how this observation is reached.

"reasonable spectrum" is replaced by

"an interpretation in which observations and theory are consistent. This is explained in what follows"

Line 251-273: The narrative discussions seem to be very qualitative. Not clear that the conclusions reached are well substantiated.

The approach employed is to assess the observations and compare those to theory to the extent the measurements allow. To that extent they are well substantiated. For example we are only able to confine propagation angles to a rather wide range, and we cannot give precise numbers, as that indeed would be unfounded in fact.

Line 275: specify that a multi-instrument data set was analyzed. Recall which data set were used.

We added as the new second sentence of the section:

"The multiple instruments used were RPC-LAP, RPC-ICA, RPC-MAG, and RPC-MIP, all part of the Rosetta Plasma Consortium."

Line 276: indicate that the waves were recorded as Langmuir probe current variations

We changed the sentence to read

"Waves which we interpret as current-driven ion acoustic waves were recorded by the Langmuir probe instrument RPC-LAP as probe current variations."

Line 283-284: The ion drift value, obtained from the analysis of the LAP I-V curve, was questioned above. What is the process causing the ion drift speed of 3 to 3.7 km/s. Could this be partly a local phenomena inside the sheath of the charged spacecraft?

Ions are accelerated by an ambipolar field radially outward from the nucleus. The sheath around the spacecraft is not affecting the measurment, as the probe is outside the sheath. In a plasma with a 2 eV electron temperature and a  $1.6 \times 10^9 \,\mathrm{m}^{-3}$  density, the Debye

length is 26 cm. The probe is therefore outside the sheath. Note that this is an upper limit. Should cold electrons be included, the resulting value would be even lower.

The Debye length estimate compared to the boom length is in the new section on uncertainties, and we added the following text about the acceleration process in the discussion section:

"Theoretical estimates of acceleration by a radial ambipolar electric field lead to ion bulk speeds in this range (Vigren and Eriksson, 2017) in agreement with observations (Odelstad et al., 2018)."

Line 285: replace "electron volt" by "eV" Changed as suggested.

Line 288: « ... possible to say something about it ». I found this statement very speculative with the limited cases tested.

We have said something about it, so it is possible. We make no claim of being able to say everything about it. The statement as it stands is not speculative.

Line 295: remove « the » before « bulk »

Changed as suggested.

Line 297: Discuss the processes that would increase the bulk temperature or form supra-thermal tails. Can wave-particle interaction contribute to the ion heating process ?

Particle populations can absorb energy through wave-particle interaction or by acceleration by dc fields, often followed by wave-particle interaction. However, it is wise not to speculate about matters that we cannot support by measurement.

**Current carried by cold electrons?**

This is part of the list of properties of the distribution that best agrees with the observations:

"It has the warmest cold ion distribution (0.04 eV), the current carried by the cold electrons, and no warm ion component as that was found to be negligible."

Line 313-316: Indeed, it seems pretty strong conclusions are reached from crude estimates and measurements with uncertainties (which are not quantified).

We think that in the amended manuscript, with the newly added discussion of the uncertainties, the conclusions are justified.